

# Reconstruction and Spatiotemporal Analysis of Global Surface Ocean pCO₂ Considering Sea Area Characteristics

Huisheng Wu, Yunlong Ji, Lejie Wang, Xiaoke Liu, Wenliang Zhou, Long Cui, Yang Wang, Min Liu and Zhuang Li

College of Oceanography and Space Informatics, China University of Petroleum (East China), Qingdao, Shandong 266580, China;

*Correspondence to:* Huisheng Wu (wuhuisheng@upc.edu.cn)

**Abstract:**

The partial pressure of carbon dioxide ($pCO_2$) on the surface of the ocean is crucial for quantifying and evaluating the ocean carbon budget. Insufficient consideration of the effects at the sea area scale makes it difficult to comprehensively evaluate the spatiotemporal distribution characteristics and variation patterns of $pCO_2$. This study constructed a $pCO_2$ evaluation dataset based on LDEO measurement data and multi-source data. After conducting correlation testing on a global, far sea, and near sea scale, a ocean surface $pCO_2$ evaluation model was

constructed using multiple linear regression, convolutional neural network, gated recurrent unit, long short-term memory network, generalized additive model, extreme gradient boosting, least squares boosting, and random forest. Performance evaluation indicates that the random-forest model consistently achieves the best accuracy across all spatial scales, yielding a global RMSE of 6.123 μatm and an $R^2$ of 0.986. In the open ocean, RMSE decreases to 4.699 μatm and $R^2$ rises to 0.988, whereas in coastal waters RMSE increases to 8.044 μatm and $R^2$

declines to 0.972. Based on this, the annual sea surface $pCO_2$ distribution of $0.25° \times 0.25°$ from 2000 to 2019 was reconstructed. The reconstructed field shows a typical equatorial high/polar low pattern, as well as an overall upward trend consistent with independent observations, with acceleration particularly evident in specific regions of subtropical coastal oceans.

**Keywords:**

Surface ocean $pCO_2$; Global Oceans; Machine learning; Spatiotemporal changes; Random forest

**Synopsis:**

This study reconstructs global ocean surface $pCO_2$ (2000–2019) using multi-source data and machine learning, identifying RF as the optimal model and revealing equatorial-high/polar-low patterns with rising trends.

## 1. Introduction

The partial pressure of carbon dioxide on the surface of the ocean ($pCO_2$) is an important indicator for measuring the exchange of $CO_2$ between the ocean and the atmosphere, and can evaluate the contribution of the ocean's carbon absorption and storage capacity to the global carbon cycle(Falkowski et al., 2000; Jain, 2022) .

Numerous scholars have conducted research on $pCO_2$ estimation and distribution reconstruction by combining satellite remote sensing data and machine learning algorithms. In the study of sea surface $pCO_2$ in local sea areas,

Telszewski et al. reconstructed the distribution of $pCO_2$ in the North Atlantic using self-organizing neural



networks(Qiu et al., 2022); Landschützer et al. reconstructed the distribution map of Atlantic sea surface pCO₂ using self-organizing map feedforward neural network method (Landschützer et al., 2013). Chierici et al. evaluated the feasibility of jointly estimating sea surface pCO₂ in Antarctica and the Pacific region using ship borne measured data and remote sensing data (Chierici et al., 2011). Nakaoka et al. established a nonlinear

relationship between sea surface pCO₂ and multiple parameters based on self-organizing neural networks, and reconstructed the spatiotemporal variation of sea surface pCO₂ in the North Pacific (Nakaoka et al., 2013). Marrec et al. used multiple linear regression to estimate the sea surface pCO₂ in the waters of the Northwest European continental shelf (Wang et al., 2021). Gregor et al. proposed methods such as support vector regression and random forest regression to reconstruct the Southern Ocean surface pCO₂ (Gregor et al., 2017); Wang et al.

reconstructed the distribution of pCO₂ on the surface of the Southern Ocean using correlation analysis and feed forward neural networks (Marrec et al., 2015). Lohrenz et al. reconstructed the sea surface pCO₂ in the northern Gulf of Mexico using regression tree algorithm (Lohrenz et al., 2021); Chen et al. compared the performance of various methods in estimating surface pCO₂ in the Gulf of Mexico (Chen et al., 2019); Fu et al. applied cubist models to estimate pCO₂ on the surface of the Gulf of Mexico (Fu et al., 2020). Zhang et al. constructed a sea

surface pCO₂ regression model for the Baltic Sea region (Zhang et al., 2021). In the study of global ocean surface pCO₂, Landschützer et al. expanded the research scope to the global level, reconstructed the pCO₂ distribution map from 1998 to 2011, and further extended it to 1982 to 2011 (Landschützer et al., 2014;Landschützer et al., 2016). Gregor et al. reconstructed the pCO₂ distribution using various nonlinear regression methods (Gregor et al., 2019). Zhong et al. used generalized regression neural network and stepwise regression algorithm to construct the

pCO₂ distribution map (Guorong et al., 2020), and combined stepwise regression algorithm and feed forward neural network, constructed a 1°× 1°pCO₂ distribution map from 1992 to 2019 according to the 11 biogeochemical provinces defined by the self-organizing map method (Zhong et al., 2020).

The aim of this study is to construct a multi-regional scale analysis framework for the global ocean, far sea areas, near sea areas, revealing the spatiotemporal variation patterns and driving mechanisms of pCO₂, and providing

scientific support for global ocean carbon sink assessment and climate change response.

## 2.   Methodology

### 2.1 Research Area

The global ocean, excluding the perennial ice-covered waters in the core area of the Arctic Ocean and the permanently frozen areas around the Antarctic continent, has a total area of 336 million square kilometers,

accounting for approximately 92.8% of the global ocean surface area. This research focuses on the 0–10-meter water layer in the ocean surface, which is a critical interface for air sea exchange. Due to the complex types of water bodies, sea surface pCO₂ is influenced by various factors. The global ocean was divided into research area scales based on water depth, identifying the areas beyond the continental shelf (water depth>200 meters) as far sea areas and the areas within the range (water depth≤200 meters) as near sea areas.

### 2.2 Data sources

### 2.2.1 Actual measurement data

The measured data of pCO₂ is sourced from Global Surface pCO₂ (LDEO) Database V2019 (OCADS - Global Surface pCO₂ (LDEO) Database (noaa. gov)). This dataset covers 14.2 million measured data from 1957 to 2019 using the equalizer CO₂ analyzer system in the global ocean. The dataset provides various types of sea surface

pCO₂ measured data. This study selected ocean surface pCO₂ values measured at actual temperatures from 2000 to 2019, which can truly reflect the pCO₂ level at the time of measurement.



### 2.2.2 Other data

A total of 25 potential influencing factors were selected for the study (Table 1), and their abbreviations are used for convenience. These data are divided into three types of sources: in-situ observations, satellite observations, and numerical models, with good spatiotemporal resolution and coverage, providing reliable data sources for research.

**Table 1. Specific information about influencing factors (sort based on its resolution and name)**

| Variable name | Abbreviation | Spatial resolution | Temporal resolution | Data type | DOI |
|---|---|---|---|---|---|
| Mass concentration of chlorophyll a in sea water | chl | 0.036 | Daily | Satellite observations | https://doi.org/10.48670/moi-00281 |
| Volume attenuation coefficient of downwelling radiative flux in sea water | kd490 | 0.036 | Daily | Satellite observations | https://doi.org/10.48670/moi-00281 |
| Ocean mixed layer thickness defined by sigma theta | mlotst* | 0.083 | Daily | Numerical models | https://doi.org/10.48670/moi-00021 |
| Sea water salinity | so | 0.083 | Daily | Numerical models | https://doi.org/10.48670/moi-00021 |
| Sea water potential temperature | thetao | 0.083 | Daily | Numerical models | https://doi.org/10.48670/moi-00021 |
| Eastward sea water velocity | uo | 0.083 | Daily | Numerical models | https://doi.org/10.48670/moi-00021 |
| Northward sea water velocity | vo | 0.083 | Daily | Numerical models | https://doi.org/10.48670/moi-00021 |
| Sea surface height above geoid | zos | 0.083 | Daily | Numerical models | https://doi.org/10.48670/moi-00021 |
| Sea surface density | dos | 0.125 | Daily | In-situ observations Satellite observations | https://doi.org/10.48670/moi-00051 |
| Sea surface salinity | sos | 0.125 | Daily | In-situ observations Satellite observations | https://doi.org/10.48670/moi-00051 |
| Mole concentration of nitrate in sea water | $no_3$ | 0.25 | Daily | Numerical models | https://doi.org/10.48670/moi-00019 |
| Mole concentration of dissolved molecular oxygen in sea water | $o_2$ | 0.25 | Daily | Numerical models | https://doi.org/10.48670/moi-00019 |
| Mole concentration of phosphate in sea water | $po_4$ | 0.25 | Daily | Numerical models | https://doi.org/10.48670/moi-00019 |
| Mole concentration of silicate in sea water | si | 0.25 | Daily | Numerical models | https://doi.org/10.48670/moi-00019 |
| Surface geostrophic eastward sea | ugos | 0.25 | Daily | Numerical | https://doi.org/10.48670/mds-003 |



| | | | | models | 27 |
|---|---|---|---|---|---|
| | | | | In-situ observations | |
| | | | | Satellite observations | |
| | | | | Numerical models | |
| Surface geostrophic northward sea water velocity | vgos | 0.25 | Daily | In-situ observations | https://doi.org/10.48670/mds-003 27 |
| | | | | Satellite observations | |
| Ocean mixed layer thickness | mlotst | 0.25 | Weekly | In-situ observations | https://doi.org/10.48670/moi-000 52 |
| | | | | Satellite observations | |
| Sea water temperature | to | 0.25 | Weekly | In-situ observations | https://doi.org/10.48670/moi-000 52 |
| | | | | Satellite observations | |
| Eastward wind | uwind | 0.25 | Monthly | Satellite observations | https://doi.org/10.48670/moi-001 81 |
| Northward wind | vwind | 0.25 | Monthly | Satellite observations | https://doi.org/10.48670/moi-001 81 |
| Aragonite saturation state in sea water | ar | 1 | Monthly | In-situ observations | https://doi.org/10.48670/moi-000 47 |
| Calcite saturation state in sea water | ca | 1 | Monthly | In-situ observations | https://doi.org/10.48670/moi-000 47 |
| Sea water ph reported on total scale | ph | 1 | Monthly | In-situ observations | https://doi.org/10.48670/moi-000 47 |
| Total alkalinity in sea water | talk | 1 | Monthly | In-situ observations | https://doi.org/10.48670/moi-000 47 |
| Dissolved inorganic carbon in sea water | $tco_2$ | 1 | Monthly | In-situ observations | https://doi.org/10.48670/moi-000 47 |

**2.3 Data Processing**

**2.3.1 Data Matching**

To reduce the impact of spatial and temporal resolution differences in multi-source data, we adopted a dual
matching strategy to process $pCO_2$ measured data and potential influencing factors. In the temporal dimension,
influencing variables were first aligned with the in-situ $pCO_2$ observations; temporal gaps were subsequently
infilled via nearest-time interpolation to ensure chronological consistency. In the spatial dimension, data points
were aligned through precise geographic coordinate matching algorithms, and nearest neighbor interpolation was
used to supplement missing points to improve spatial accuracy. After matching, each point contains the measured
value of $pCO_2$, environmental variables, and corresponding spatiotemporal information (year, month, lat, lon).

**2.3.2 Analysis of Outliers**



The study conducted quality control on the matched data by removing missing values generated during the matching process. According to data statistics and previous research experience (19), measured data below 200μatm and above 600μatm are classified as outliers. The spatial distribution of outliers is mainly concentrated in coastal areas, reflecting the variability of land sea interaction effects. Outliers are valuable sample data for the study of $pCO_2$. Through comparative analysis of each route, it was found that many outliers matched the route, and it was determined that their outliers were caused by environmental changes rather than measurement errors. Therefore, valid outliers were retained and only obvious measurement error data were removed. For other environmental variable values, abnormal data was identified and removed based on the 3σcriterion (μ±3σ).

### 2.3.3 Data Balancing

The processed global ocean data was divided into far sea and near sea datasets (Figure 1a, b, c). Statistical analysis shows that the spatial and temporal distribution of data is uneven. Therefore, a 0.25°× 0.25°grid was used for spatial binning, and time binning was performed monthly to construct a spatiotemporal joint binning unit. The granularity setting of this box not only meets the research accuracy requirements, but also maintains compatibility with the spatiotemporal resolution of multi-source data.

Take the arithmetic mean of the data within each unit as the representative value, with the spatial position represented by the grid center point, and the time calculated as the weighted average based on the distribution of data points (Formula 1). This method effectively balances the data distribution while ensuring accuracy.

$$t_{avg} = \frac{\sum_{i=1}^{n} w_i t_i}{\sum_{i=1}^{n} w_i} \qquad (1)$$

$$w_i = \Delta t_i \qquad (2)$$

In the formula, $t_{avg}$ is the weighted average time of the spatiotemporal box, n is the total amount of data in the spatiotemporal box, $w_i$ is the weight of the i-th data point, $t_i$ is the time of the i-th data point, and $\Delta t_i$ is the sampling time interval between the i-th data point and the previous point. After data balancing processing, the dataset for this study was finally constructed, laying a solid data foundation for the construction of multi-scale models.

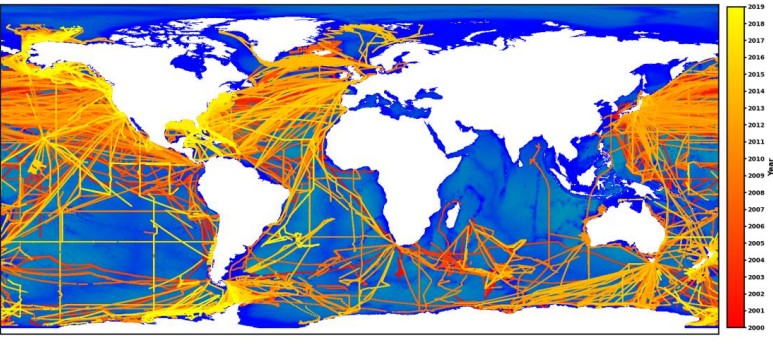

**(a)**



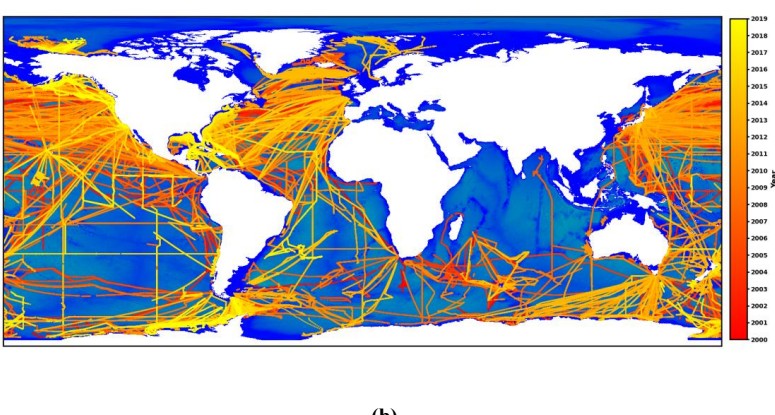

**(b)**

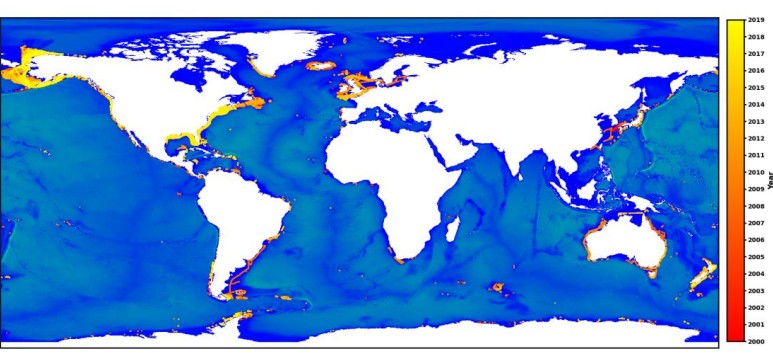

**(c)**

**Figure 1. The spatiotemporal distribution of datasets at different scales.(a)Global spatial distribution of ocean data.(b)Spatial distribution of data in far sea areas.(c)Spatial distribution of data in near sea areas.**

**2.4 Spearman correlation analysis of pCO₂ drivers**

The potential influencing factors involved do not fully follow a normal distribution, and there is a non-linear relationship between pCO₂. Therefore, selecting appropriate correlation indicators is particularly crucial. The Spearman correlation coefficient can effectively reveal the correlation between data (Formula 3).

$$\rho = 1 - \frac{6\sum_{i=1}^{n} D_i^2}{n(n^2-1)} \tag{3}$$

In the formula, $\rho$ represents the correlation coefficient, $d$ represents the level difference of the variable, and $n$ represents the sample size of the variable. The range of values for $\rho$ is between -1 and 1, where -1 indicates a complete negative correlation between the influencing factors and pCO₂, 1 indicates a complete positive correlation, and 0 indicates no correlation.

**2.5 Model selection**



To evaluate the modeling ability of different algorithms for $pCO_2$, we constructed eight comparative models at different research regions, including multiple linear regression (MLR),convolutional neural network (CNN), gated recurrent unit (GRU), long short term memory (LSTM),generalized additive models (GAM), extreme gradient boosting (XGBoost), least squares boosting (LSBoost), and random forest (RF). MLR serves as a baseline that linearly links temperature, salinity and nutrients to sea-surface $pCO_2$. CNN extracts spatial features

via convolution and pooling layers to produce fine-scale $pCO_2$ distributions, while GRU and LSTM, with their update-reset gates and memory cells, capture long-term temporal dependencies of oceanic periodic changes on $pCO_2$ for historical-to-future prediction. GAM relaxes the linearity assumption by modeling each predictor's additive nonlinear effect on $pCO_2$. XGBoost and LSBoost iteratively optimize tree ensembles through gradient boosting or weighted residuals to uncover complex nonlinear relationships between high-dimensional features

and $pCO_2$. Finally, RF constructs and averages many decision trees on random feature subsets, delivering robust $pCO_2$ estimates for large-scale ocean datasets.

**2.6 Performance evaluation**

    The datasets at different research regions were randomly divided into training, validation, and testing sets in an 8:1:1 ratio. Five statistical methods, Mean Absolute Error (MAE, µatm) – the average absolute difference

between predicted and in-situ $pCO_2$, indicating overall bias; Mean Absolute Percentage Error (MAPE, %)–the relative error scaled by the observed $pCO_2$, enabling comparison across regions with contrasting background concentrations; Mean Squared Error (MSE, µatm²) – the squared deviations averaged over all samples, emphasizing larger $pCO_2$ discrepancies; Root Mean Squared Error (RMSE, µatm) – the square root of MSE, providing a metric in the original $pCO_2$ units that is sensitive to outliers; Coefficient of Determination ($R^2$) – the

proportion of $pCO_2$ variance explained by the model, with values approaching unity signifying high predictive skill.

$$MAE = \frac{1}{n} \sum_{i=1}^{n} \left| \hat{y}_i - y_i \right| \tag{4}$$

$$MAPE = \frac{100\%}{n} \sum_{i=1}^{n} \left| \frac{\hat{y}_i - y_i}{y_i} \right| \tag{5}$$

$$MSE = \frac{1}{n} \sum_{i=1}^{n} \left( \hat{y}_i - y_i \right)^2 \tag{6}$$

$$RMSE = \sqrt{\frac{1}{n} \sum_{i=1}^{n} \left( \hat{y}_i - y_i \right)^2} \tag{7}$$

$$R^2 = 1 - \frac{\sum_{i=1}^{n} \left( y_i - \hat{y}_i \right)^2}{\sum_{i=1}^{n} \left( y_i - \bar{y}_i \right)^2} \tag{8}$$

    In the formula, n is the number of $pCO_2$ observations; $y_i$ denotes the in-situ measured $pCO_2$ (µatm) for the i-th sample, $\hat{y}_i$ is the corresponding model-estimated $pCO_2$, $\bar{y}_i$ represents the mean of all measured $pCO_2$ values.

**3.    Results and discussion**

**3.1 Correlation detection**

**3.1.1 Interaction detection**

    Interactive detection of variables was conducted in global oceans, far sea areas, and near sea areas (Figure 2). The concentration of chlorophyll and the volume attenuation coefficient of downwelling radiative flux have a p-value of 1 at all research area scales, indicating collinearity in numerical values. However, they respectively reflect





marine biological activity and optical properties, providing comprehensive information for fitting surface pCO₂. The ρ value between the aragonite saturation state in sea water and aragonite in seawater is also 1, and they are positively correlated with the same magnitude of change. This usually stems from chemical equilibrium processes in seawater, where the dissolution and precipitation processes are influenced by similar physical and chemical conditions. The correlation between sea water potential temperature and sea water temperature is extremely high,

but their physical meanings are different. The former reflects the equivalent temperature after considering pressure, while the latter reflects the actual temperature. Both can comprehensively capture temperature characteristics and improve the accuracy of surface pCO₂ evaluation.

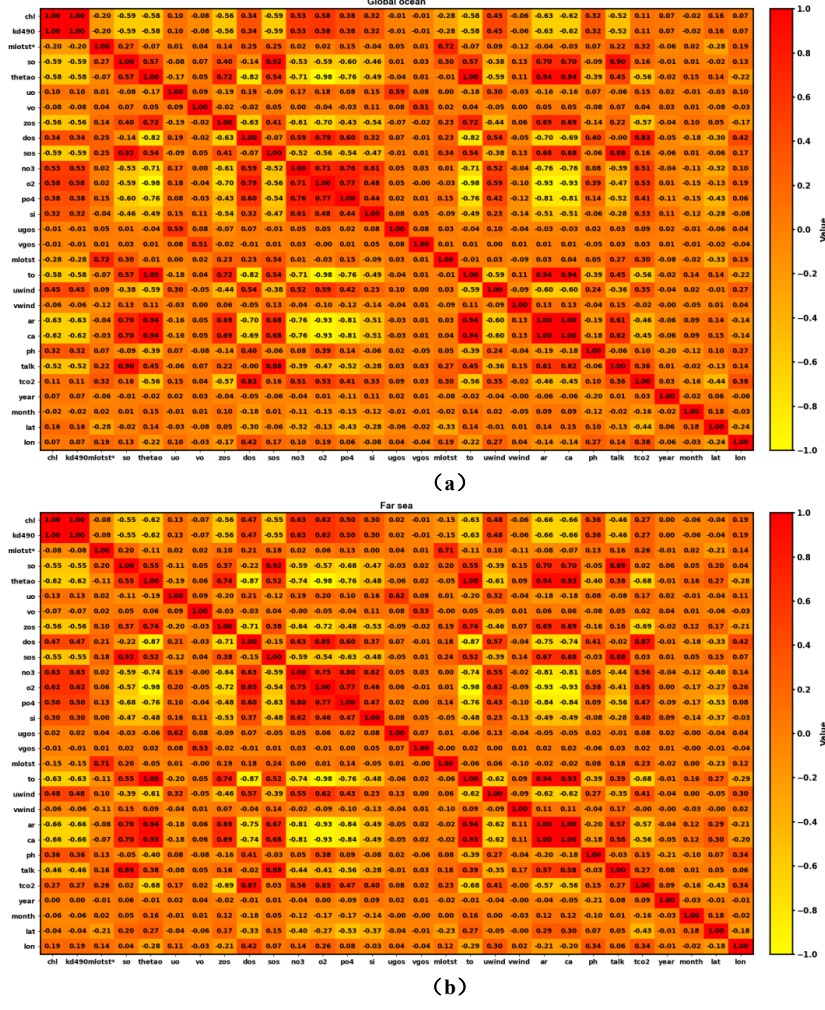





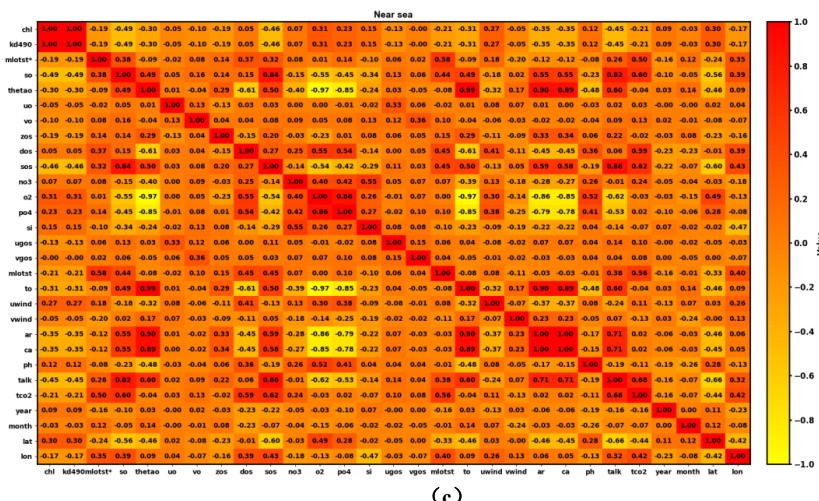

(c)

**Figure 2. Results of interaction detection between variables at different research area scales**.(a)**Global Ocean Interaction Detection Results.(b)Interaction detection results in far sea areas.(c)Interactive detection results in near sea areas.**

### 3.1.2 Single factor detection

The correlation between surface pCO₂ and various influencing factors (Figure 3) was analyzed. The results indicate that at different regional scales, there is a significant negative correlation between pCO₂ and ph, meaning that the stronger the acidity of seawater, the higher the surface pCO₂; the stronger the alkalinity, the lower the surface pCO₂. At the same time, surface pCO₂ is significantly positively correlated with temperature. In far sea areas, the negative correlation between pCO₂ and chlorophyll concentration and diffuse reflectance attenuation coefficient is more significant, indicating that it has higher stability and balance in regulating pCO₂. In contrast, the above correlation in near sea areas is weaker due to land-based pollution, human activities, and environmental changes, but the negative correlation between pCO₂ and seawater acidity is stronger. When selecting variables, the study included factors with a p-value greater than 0.1 or less than -0.1 in the screening range to ensure the validity of the results and improve model performance (Table 2). Additionally, SHAP method was used to quantitatively evaluate the contributions of various influencing factors to surface pCO₂ (20). There were differences in the contributions of influencing factors at different scales. The ph is the core driving factor at all scales, but its contribution intensity follows a distribution pattern of "far sea areas>global oceans>near sea areas"; The contribution of other factors shows significant regional heterogeneity, such as talk being the second key factor at the global ocean scale, while the contribution rate of o₂ in near sea areas has significantly increased, making ar a region specific factor.

**Table 2. Selection results of influencing factors at different area scales**

| Research scale | Influence factor |
|---|---|
| Global Ocean | ph、o₂、chl、kd490、dos、uwind、po₄、lon、zos、month、sos、year、talk、ca、so、ar、to、thetao |
| Far sea | ph、chl、kd490、o₂、dos、lon、uwind、po₄、zos、month、sos、talk、so、ca、ar、year、to、thetao |
| Near sea | ph、o₂、po₄、lat、dos、no₃、chl、kd490、mlotst*、tco₂、lon、 month、ca、ar、sos、 |



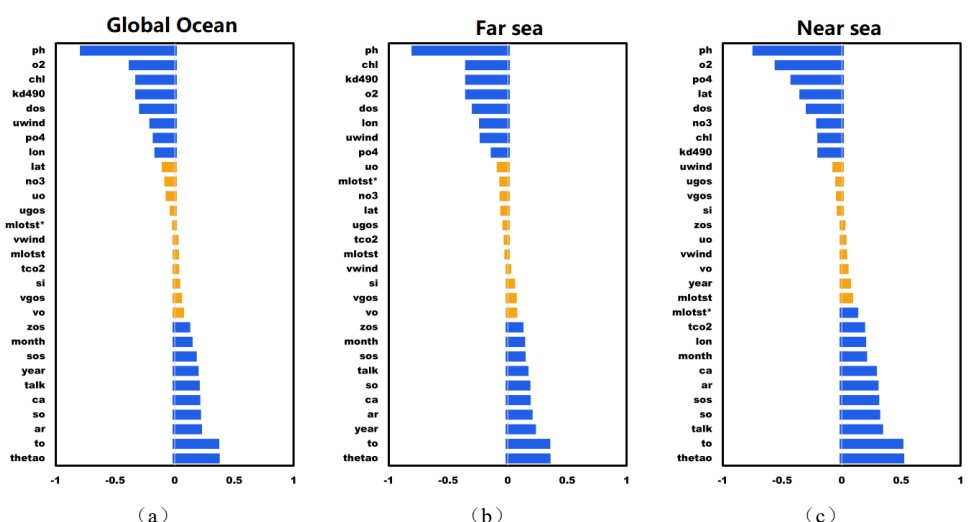

**Figure 3. Single factor detection results at different research area scales**.(a)**Global ocean single factor detection results.(b)Far sea single factor detection results.(c)Near sea single factor detection results**

### 3.2 Model construction and evaluation

#### 3.2.1 Construction and evaluation of global ocean surface pCO₂ model

Different models exhibit significant performance differences in evaluating surface pCO₂ at the global ocean scale (Figure 4). Specifically, there is a significant gap between the model values of MLR, CNN, and GRU and the true values, especially in the low value (<300μatm) and high value (>500μatm) ranges where the fitting effect is poor (Table 3). The deviation is due to the model's insufficient ability to capture nonlinear relationships in complex marine environments, limitations in handling extreme values, and the model's own structure is not sufficient to adapt to complex data features. The LSTM and GAM models have relatively large errors and poor performance, indicating deficiencies in capturing the characteristics of surface pCO₂ changes. When extreme fluctuations occur in surface pCO₂, the fitting ability significantly decreases. The comprehensive performance of XGBoost and LSBoost has significantly improved, with MAE reduced to 15μatm~18μatm, RMSE reduced to 25μatm~30μatm, and R² exceeding 0.7. The effective explanation of multivariate nonlinear relationships and the application of model ensemble strategies have improved the accuracy of the two models within the normal range (300μatm~500μatm), but the extreme values processing still needs to be improved. The performance of RF is the best among all models, with MAE reduced to below 4μatm, RMSE reduced to around 6μatm, and R² reaching above 0.9. It not only achieves accurate fitting in the range of 300μatm~500μatm values, but also in the low and high value ranges. The good adaptability of RF to high-dimensional data and a large number of samples makes it perform well in fitting tasks in complex marine environments.

**Table 3. Performance parameters of different models in the global ocean**

|          | Model | MAE/μatm | MAPE | MSE/μatm² | RMSE/μatm | R² |
|----------|-------|----------|------|-----------|-----------|-----|
| Training | RF | 3.895 | 0.011 | 46.162 | 6.794 | 0.983 |
|          | LSBoost | 15.626 | 0.045 | 664.186 | 25.772 | 0.783 |





|  |  |  |  |  |  |  |
|---|---|---|---|---|---|---|
|  | XGBoost | 17.262 | 0.051 | 908.879 | 30.148 | 0.703 |
|  | GAM | 19.903 | 0.058 | 1398.695 | 37.399 | 0.543 |
|  | LSTM | 18.664 | 0.055 | 1430.072 | 37.816 | 0.533 |
|  | GRU | 19.053 | 0.056 | 1480.157 | 38.473 | 0.516 |
|  | CNN | 19.903 | 0.058 | 1484.621 | 38.531 | 0.515 |
|  | MLR | 19.952 | 0.058 | 1615.155 | 40.189 | 0.472 |
| Validation | RF | 3.902 | 0.011 | 46.099 | 6.790 | 0.983 |
|  | LSBoost | 15.604 | 0.045 | 661.203 | 25.714 | 0.788 |
|  | XGBoost | 17.255 | 0.051 | 910.387 | 30.173 | 0.708 |
|  | GAM | 19.905 | 0.058 | 1429.372 | 37.807 | 0.541 |
|  | LSTM | 18.675 | 0.055 | 1463.378 | 38.254 | 0.529 |
|  | GRU | 19.059 | 0.056 | 1515.286 | 38.927 | 0.513 |
|  | CNN | 19.901 | 0.058 | 1520.882 | 38.999 | 0.511 |
|  | MLR | 19.969 | 0.058 | 1656.093 | 40.695 | 0.468 |
| Testing | RF | 3.697 | 0.010 | 37.485 | 6.123 | 0.986 |
|  | LSBoost | 15.602 | 0.045 | 660.401 | 25.698 | 0.785 |
|  | XGBoost | 17.284 | 0.051 | 914.165 | 30.235 | 0.703 |
|  | GAM | 19.916 | 0.058 | 1399.851 | 37.415 | 0.545 |
|  | LSTM | 18.690 | 0.055 | 1431.489 | 37.835 | 0.535 |
|  | GRU | 19.079 | 0.056 | 1483.044 | 38.510 | 0.518 |
|  | CNN | 19.927 | 0.058 | 1488.331 | 38.579 | 0.516 |
|  | MLR | 19.982 | 0.058 | 1621.378 | 40.266 | 0.473 |

(a)MLR

(b)CNN



(c)GRU

(d)LSTM

(e)GAM

(f)XGBoost

(g)LSBoost




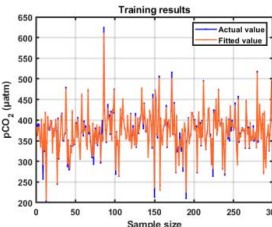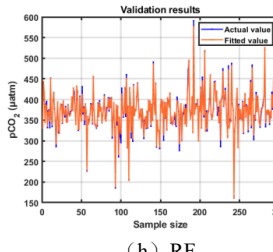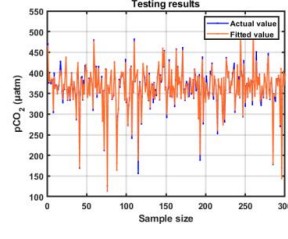

(h)RF

**Figure 4. Model performance at the global ocean**

### 3.2.2 Construction and evaluation of surface pCO₂ model in far sea areas

The far sea environment is relatively stable, and the model performance has been improved (Table 4). The bias of MLR, CNN, and GRU models has been reduced, with MAE ranging from 14μatm to 15μatm, RMSE above 26μatm, and $R^2$ remaining around 0.6. The MAE of LSTM and GAM is around 14μatm; RMSE is above 25μatm, and $R^2$ is around 0.64. The performance of the two models has improved compared to extreme value ranges, thanks to the ability of LSTM to process time series data and capture the dynamic characteristics of surface pCO₂ over time, and GAM fitted the relationship between surface pCO₂ and influencing factors by constructing a nonlinear additive model. XGBoost and LSBoost perform even better in far sea areas, especially with high fitting accuracy in the range of 300μatm~500μatm, MAE around 11μatm~13μatm, RMSE reduced to below 23μatm, and $R^2$ increased to around 0.8. The model performance of RF in far sea areas is also optimal, relying on strong generalization ability and feature selection mechanisms to effectively address the variability factors in marine environments.

**Table 4. Performance parameters of different models in the far sea areas**

| | Model | MAE/μatm | MAPE | MSE/μatm² | RMSE/μatm | $R^2$ |
|---|---|---|---|---|---|---|
| Training | RF | 3.068 | 0.009 | 27.456 | 5.240 | 0.985 |
| | LSBoost | 11.509 | 0.033 | 337.852 | 18.381 | 0.813 |
| | XGBoost | 13.191 | 0.038 | 500.054 | 22.362 | 0.723 |
| | GAM | 14.066 | 0.040 | 623.501 | 24.970 | 0.654 |
| | LSTM | 14.160 | 0.041 | 647.853 | 25.453 | 0.641 |
| | GRU | 14.377 | 0.041 | 665.920 | 25.805 | 0.631 |
| | CNN | 14.882 | 0.043 | 681.120 | 26.098 | 0.623 |
| | MLR | 15.274 | 0.044 | 737.902 | 27.164 | 0.591 |
| Validation | RF | 3.061 | 0.009 | 27.110 | 5.207 | 0.985 |
| | LSBoost | 11.532 | 0.032 | 338.102 | 18.388 | 0.814 |
| | XGBoost | 13.243 | 0.038 | 511.318 | 22.612 | 0.719 |
| | GAM | 14.143 | 0.040 | 644.144 | 25.380 | 0.646 |
| | LSTM | 14.219 | 0.040 | 667.947 | 25.845 | 0.632 |
| | GRU | 14.441 | 0.041 | 686.351 | 26.198 | 0.622 |
| | CNN | 14.929 | 0.042 | 701.278 | 26.482 | 0.614 |
| | MLR | 15.336 | 0.043 | 758.818 | 27.547 | 0.582 |
| Testing | RF | 2.900 | 0.008 | 22.082 | 4.699 | 0.988 |
| | LSBoost | 11.521 | 0.032 | 339.772 | 18.433 | 0.813 |
| | XGBoost | 13.223 | 0.038 | 508.771 | 22.556 | 0.720 |



| | MAE | MAPE | MSE | RMSE | R² |
|---|---|---|---|---|---|
| GAM | 14.104 | 0.040 | 638.362 | 25.266 | 0.649 |
| LSTM | 14.201 | 0.040 | 663.510 | 25.759 | 0.635 |
| GRU | 14.423 | 0.041 | 681.866 | 26.113 | 0.625 |
| CNN | 14.914 | 0.042 | 696.718 | 26.395 | 0.617 |
| MLR | 15.316 | 0.043 | 754.142 | 27.462 | 0.585 |

### 3.2.3 Construction and evaluation of surface pCO₂ model in near sea areas

Due to various complex factors, the spatiotemporal distribution of surface pCO₂ in the near sea area exhibits high variability, resulting in a decrease in the performance of the model. Table 5 results show that MLR, CNN, and GRU have limitations in handling complex nonlinear relationships. In the low and high value ranges, the MAE of the three models reaches over 34μatm, RMSE reaches over 62μatm, and R² is below 0.5. LSTM constructs a nonlinear additive model through its gating mechanism and GAM, which improves the fitting ability to a certain extent. The MAE of the model is in the range of 33μatm~34μatm; the RMSE is in the range of 56μatm~58μatm, and the R² remains in the range of 0.55~0.60, but there is still deviation in the extreme numerical range. XGBoost and LSBoost improved the accuracy of fitting extreme values by constructing multiple weak learners to combine the fitting results. The MAE of both models decreased to around 23μatm~27μatm, the RMSE remained around 35μatm~42μatm, and the R²increased to the range of 0.75~0.85. RF constructed multiple decision trees and integrated the fitting results to adapt to the variability and variability of the near sea environment, demonstrating robust fitting performance. Its MAE was below 5μatm; RMSE was about 8μatm, and R² remained above 0.95, significantly outperforming other models.

**Table 5. Performance parameters of different models in the near sea areas**

| | Model | MAE/μatm | MAPE | MSE/μatm² | RMSE/μatm | R² |
|---|---|---|---|---|---|---|
| Training | RF | 5.396 | 0.016 | 98.332 | 9.916 | 0.977 |
| | LSBoost | 23.673 | 0.071 | 1267.869 | 35.607 | 0.833 |
| | XGBoost | 27.298 | 0.083 | 1783.422 | 42.231 | 0.765 |
| | GAM | 34.088 | 0.102 | 3058.776 | 55.306 | 0.597 |
| | LSTM | 32.738 | 0.100 | 3273.977 | 57.219 | 0.569 |
| | GRU | 34.022 | 0.103 | 3754.637 | 61.275 | 0.505 |
| | CNN | 36.309 | 0.110 | 3989.599 | 63.163 | 0.474 |
| | MLR | 36.264 | 0.109 | 4426.775 | 66.534 | 0.417 |
| Validation | RF | 5.346 | 0.016 | 93.028 | 9.645 | 0.978 |
| | LSBoost | 23.604 | 0.071 | 1263.495 | 35.546 | 0.832 |
| | XGBoost | 27.234 | 0.083 | 1766.706 | 42.032 | 0.765 |
| | GAM | 34.040 | 0.102 | 3033.228 | 55.075 | 0.596 |
| | LSTM | 32.686 | 0.100 | 3259.080 | 57.088 | 0.566 |
| | GRU | 33.987 | 0.103 | 3727.152 | 61.050 | 0.504 |
| | CNN | 36.239 | 0.110 | 3955.729 | 62.895 | 0.474 |
| | MLR | 36.188 | 0.109 | 4387.955 | 66.242 | 0.416 |
| Testing | RF | 4.756 | 0.014 | 64.708 | 8.044 | 0.972 |
| | LSBoost | 23.564 | 0.071 | 1244.921 | 35.283 | 0.839 |
| | XGBoost | 27.299 | 0.083 | 1788.363 | 42.289 | 0.769 |
| | GAM | 34.204 | 0.102 | 3134.086 | 55.983 | 0.595 |
| | LSTM | 32.911 | 0.100 | 3394.342 | 58.261 | 0.562 |





| | | | | | |
|---|---|---|---|---|---|
| GRU | 34.236 | 0.103 | 3904.309 | 62.485 | 0.496 |
| CNN | 36.465 | 0.110 | 4132.316 | 64.283 | 0.466 |
| MLR | 36.405 | 0.109 | 4594.537 | 67.783 | 0.406 |

### 3.3 Independent validation of the model

The surface $pCO_2$ models were independently validated at different regional scales, inputting data independent of
the model construction, comparing the accuracy of the fitted values with the true values, and evaluating the
applicability and accuracy of the model in complex marine environments. The scatter plot with true values as the
x-axis and fitted values as the y-axis was drawn, with colors representing kernel density to reflect the distribution
trend of points. At the global ocean scale (Figure 5), the scatter distribution of MLR, CNN, GRU, LSTM, and
GAM shows a large elliptical shape, and the fitted values deviate significantly from the true values, especially
around the extreme value of $pCO_2$ on the sea surface. The scatter distributions of XGBoost and LSBoost have
shrunk. The RF model has the best fitting performance, with a clear convergence of the scatter distribution,
concentrated on Y=X line, and can effectively avoid errors in the extreme value region, indicating that its fitted
value is consistent with the true value and has good stability.

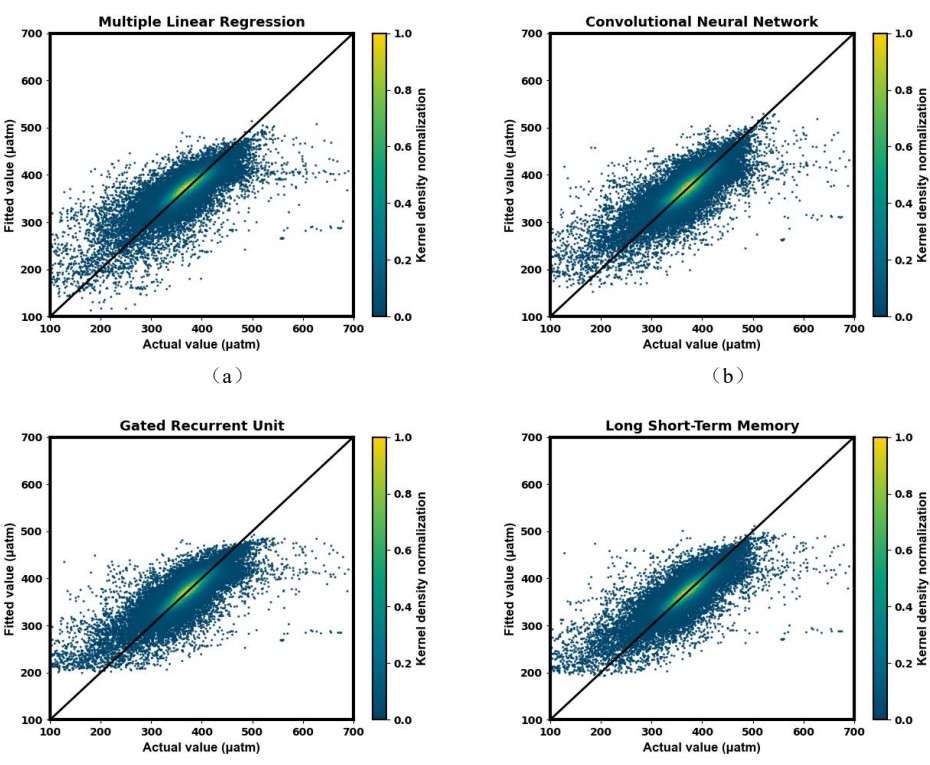



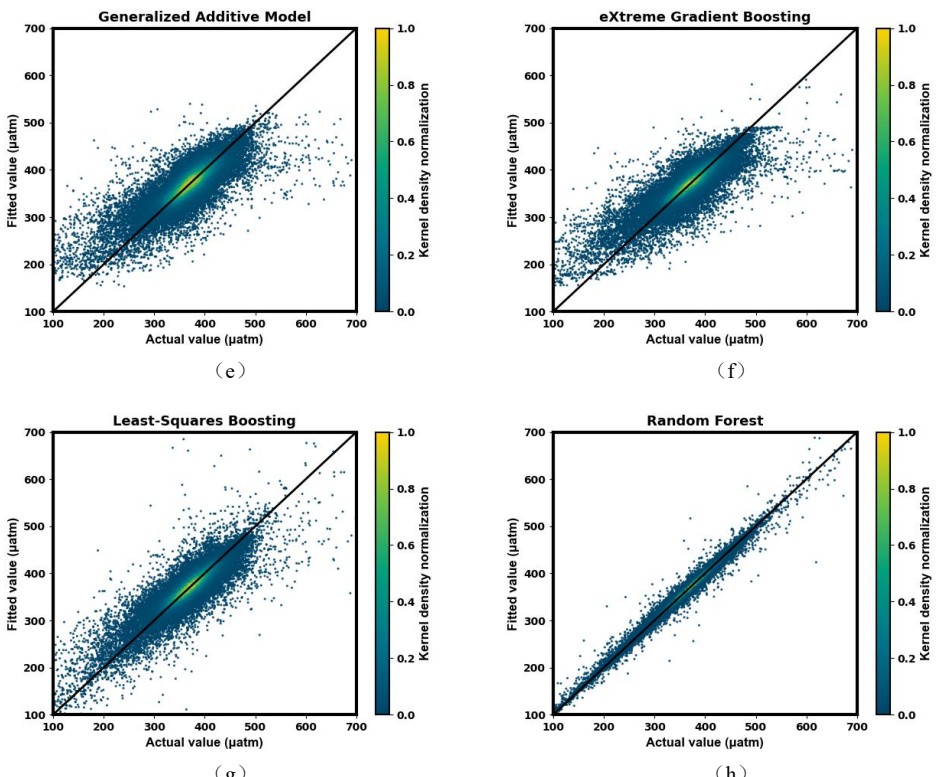

**Figure 5. Independent verification performance of the models in the global ocean, right axis: Normalized probability density of model residuals.((a)MLR,(b)CNN,(c)GRU,(d)LSTM,(e)GAM,(f)XGBoost,(g)LSBoost,(h)RF)**

In far sea areas (Figure 6), the scatter points of MLR, CNN, GRU, LSTM, GAM, and XGBoost models exhibit elliptical distribution and diverge at both ends, indicating their limitations in dealing with extreme fluctuations of surface $pCO_2$. The scatter distribution ellipse of the LSBoost model significantly shrinks, and the divergence situation converges at extreme values, improving the fitting accuracy. The scatter distribution of the RF model is a flat ellipse, with the minimum difference between the fitted value and the true value, effectively reducing extreme errors.



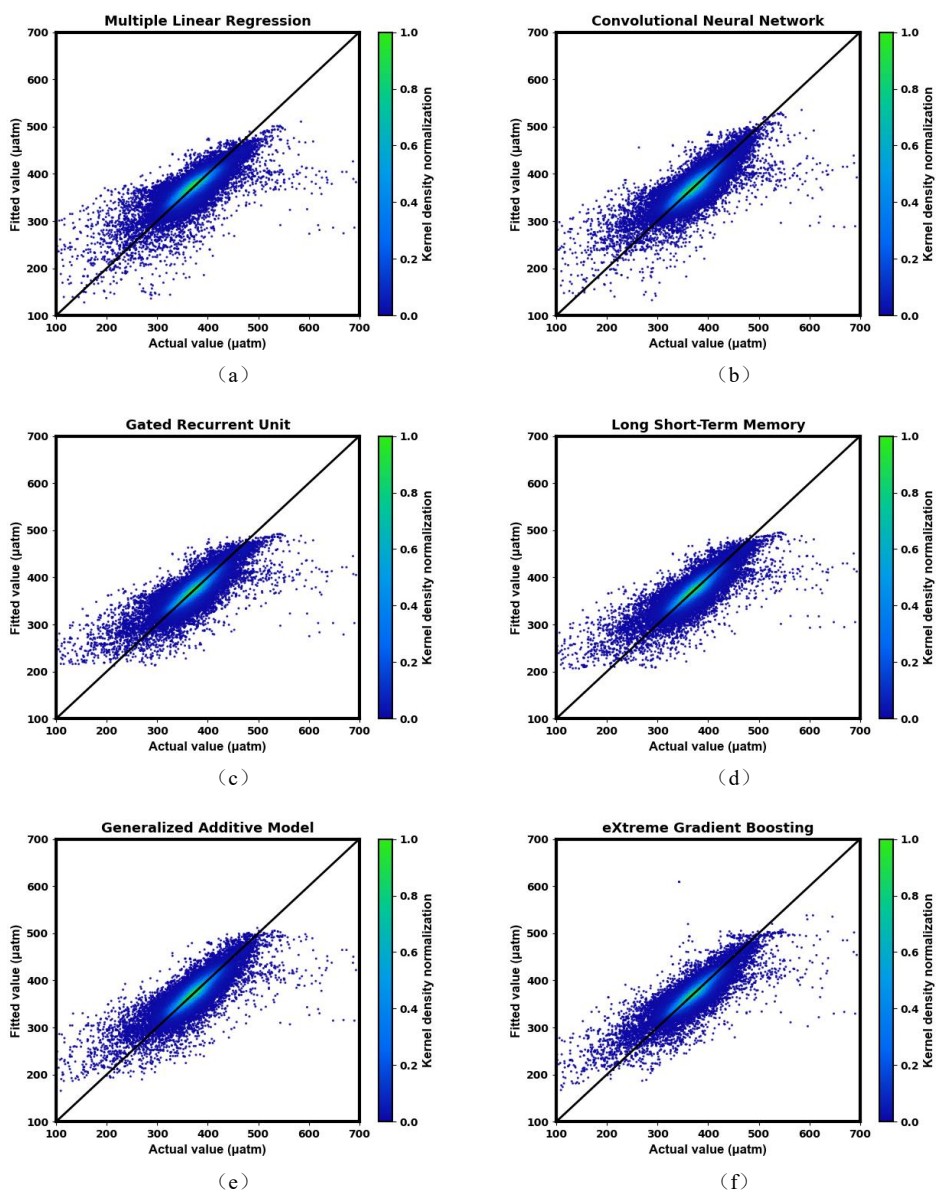



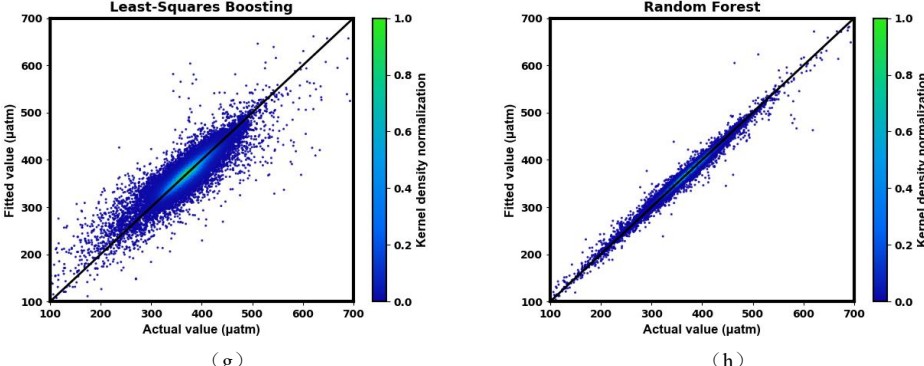

(g) (h)

**Figure 6. Independent verification performance of the models in the far sea areas, right axis: Normalized probability density of model residuals.((a)MLR,(b)CNN,(c)GRU,(d)LSTM,(e)GAM,(f)XGBoost,(g)LSBoost,(h)RF)**

In the independent validation of models in near sea areas, each model showed different performances (Figure 7).
The scatter of MLR, CNN, GRU, and LSTM shows an irregular distribution, with significant differences between the fitted values and the true values, and severe divergence in high-value areas. This is due to the high variability in near sea areas, which makes it difficult for the model to cope with. The scatter distribution of GAM and XGBoost has begun to show an elliptical shape, which has certain adaptability to complex environments. The scatter distribution of LSBoost shows a clear elliptical shape, which improves the fitting stability. The RF model shows
significant improvement in performance, with overall convergence of scatter distribution and no significant divergence in both low and high value oceans. It can effectively reduce extreme errors and reconstruct surface pCO₂ with high accuracy in complex near sea environments.

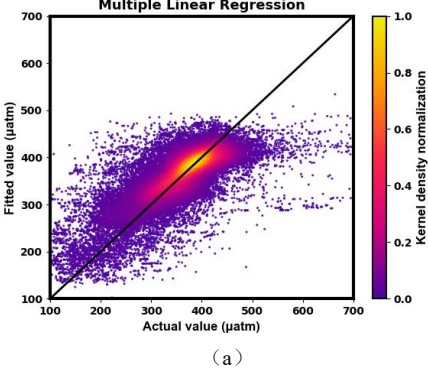 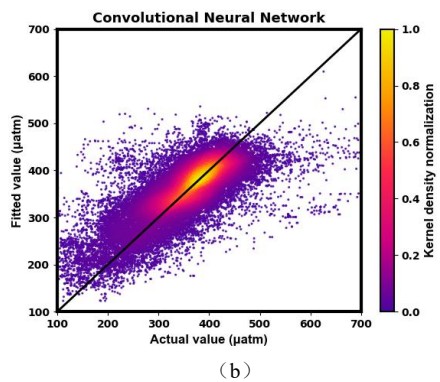

(a) (b)



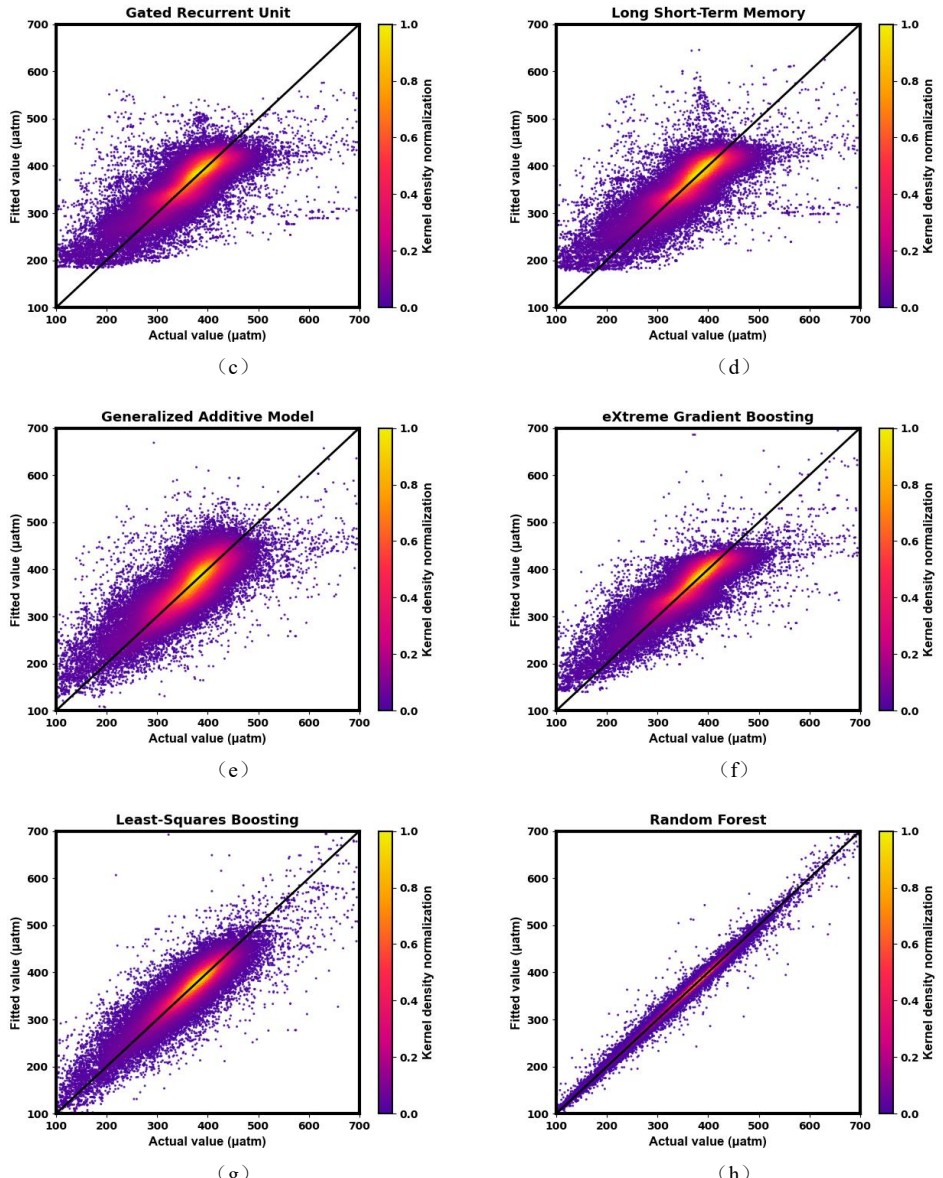

**Figure 7. Independent verification performance of the models in the near sea areas, right axis: Normalized probability density of model residuals.((a)MLR,(b)CNN,(c)GRU,(d)LSTM,(e)GAM,(f) XGBoost,(g)LSBoost,(h)RF)**

### 3.4 Reconstruction of surface pCO₂

The multi-source data was input into the constructed RF model at different area scales, with extracting the variable values of influencing factors from the multi-source data grid by grid to fit the surface $pCO_2$ values of the corresponding grid. If there are missing values in a certain grid in the multi-source data, the corresponding surface $pCO_2$ value at that location will be output as a blank value, ensuring that the reconstructed results are completely



based on the original data. The final generation of the surface pCO₂ distribution map for the year 2000~2019 at 0.25°× 0.25°is based on the original data.

The reconstruction results of surface pCO₂ at the global ocean scale are consistent with the distribution characteristics of LDEO actual observation data, confirming that the RF model can effectively capture the spatial distribution pattern of global ocean surface pCO₂. Through the reconstruction results (Figure 8), it was found that the spatial distribution of surface pCO₂ exhibits a clear latitude dependence, with a distribution pattern of "high at the equator and low at the poles". The independent observation data based on the route was compared with the reconstruction results obtained at the closest collection time. The global ocean surface pCO₂ reconstruction result showed MAE of 11.067μatm, MAPE of 0.037, MSE of 396.060μatm², RMSE of 19.901μatm, and $R^2$ of 0.816. This indicates that the deviation between the reconstructed results and the actual observed data is small, and can accurately reflect the average distribution characteristics of surface pCO₂.

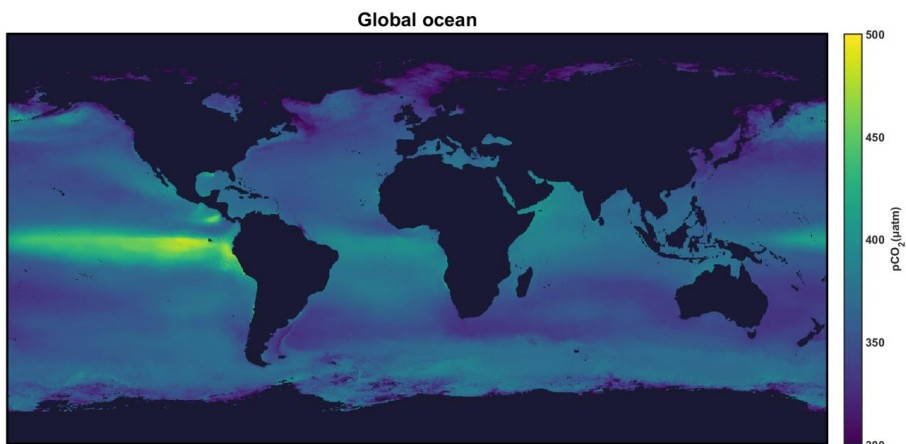

**Figure 8. Surface ocean pCO₂ products in the global ocean**

Compared with other existing studies on the reconstruction of surface pCO₂, these methods are highly consistent with our results in the reconstructed spatial model pattern(Zhong et al., 2022; Chau et al., 2021; Chau et al., 2022). Although different studies have used different data sources, models, or methods, similar conclusions can be drawn when describing the overall distribution characteristics of pCO₂ on the global ocean surface, which to some extent verifies the reliability and accuracy of the reconstructed results. This study uses high-resolution data and RF models to make the reconstruction results more detailed, especially in the high latitude marginal sea areas of the North and South Poles.

The reconstruction results of the far sea region showed that the surface pCO₂ in the equatorial low latitude region was higher, while the surface pCO₂ in the polar high latitude region was lower (Figure 9). We evaluated the difference in fitting accuracy between the far sea regional model and the global ocean model in the far sea areas, by comparing independent observation data based on flight routes with the reconstructed results of the two models. The results showed that the MAE of the far-sea model was 9.060 μatm, the MAPE was 0.027, the MSE was 269.511 μatm², the RMSE was 16.417 μatm, and $R^2$ was 0.826; the MAE of the global model was 9.125 μatm, the MAPE was 0.027, the MSE was 275.582 μatm², the RMSE was 16.601 μatm, and $R^2$ was 0.822.The reconstruction accuracy of the far sea area model has slightly improved compared to the global ocean model in the far sea area (Figure 10), indicating that the optimization of the far sea area model in local areas has improved the




reconstruction accuracy. However, the global ocean model can still provide accurate surface $pCO_2$ fitting in the far sea area by adapting to the overall ocean environment.

To verify the accuracy of the time series reconstruction of the model, a comparative analysis was conducted on the temporal changes between the observation data of the Hawaii Ocean Time series (HOT) and the reconstruction results of the global ocean and far sea areas (Figure 11). The results showed that the temporal trends of both scales

were consistent with the actual measurement data of the Hawaii observation station. Research has shown that the model performs well in fitting the dynamic changes of time series and can accurately reflect the temporal evolution of surface $pCO_2$.

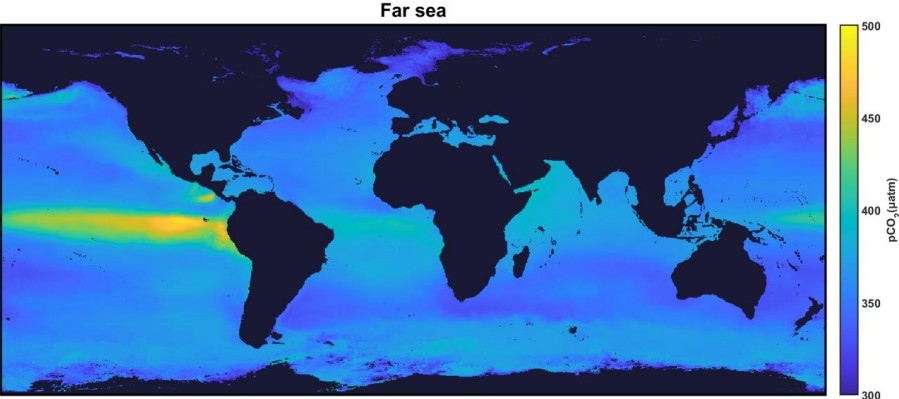

**Figure 9. Surface ocean pCO₂ products in the far sea areas**

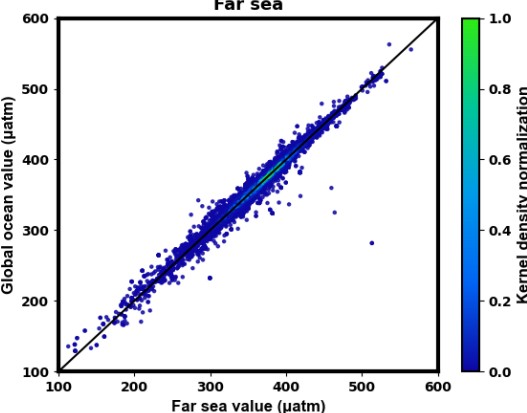

**Figure 10. Comparison of reconstruction accuracy in the far sea areas using different scale models, right axis: Normalized probability density of model residuals.**




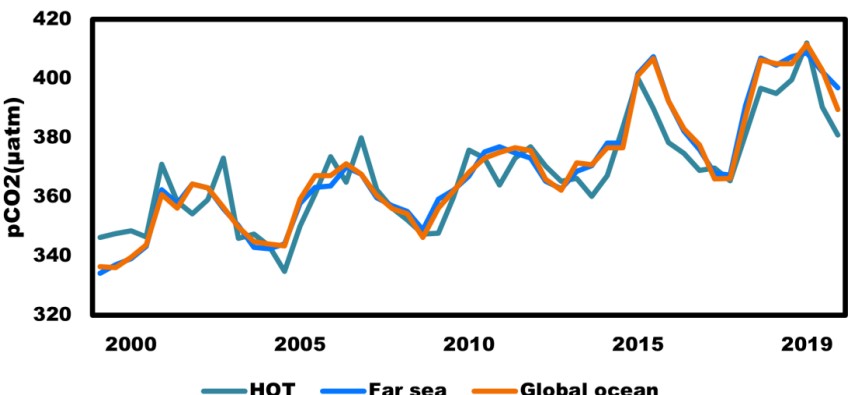

**Figure 11. Independent verification based on time-series observation stations**

The reconstruction results of surface pCO₂ in the near sea area showed (Figure 12) that the surface pCO₂ values in the low latitude near sea areas on both sides of the equator were higher, which was closely related to factors such as high seawater temperature and vigorous evaporation. The seawater temperature in high latitude oceans is lower, causing changes in ocean circulation and mixing processes, and the overall trend of surface pCO₂ is decreasing. A comparison was made between the fitting accuracy of the near sea area model and the global ocean model in the

near sea region. The results showed that the MAE of the near-shore model was 20.145 μatm, the MAPE was 0.065, the MSE was 983.726 μatm², the RMSE was 31.364 μatm, and R² was 0.797; the MAE of the global model was 20.324 μatm, the MAPE was 0.065, the MSE was 999.147 μatm², the RMSE was 31.609 μatm, and R² was 0.794. The reconstruction effect of the near sea area model has been improved compared to the reconstruction results of the global ocean model in the near sea area (Figure 13), indicating that the use of RF can model the complex

marine environment in the near sea area and accurately reflect the distribution characteristics of surface pCO₂ in the region.

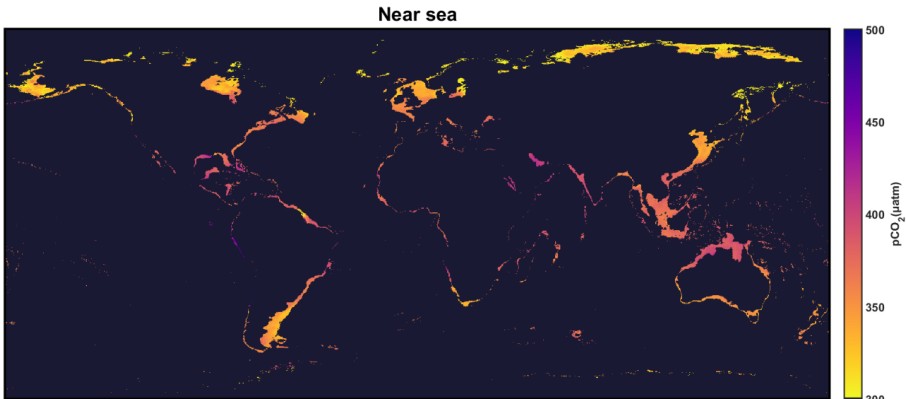

**Figure 12. Surface ocean pCO₂ products in the near sea areas**





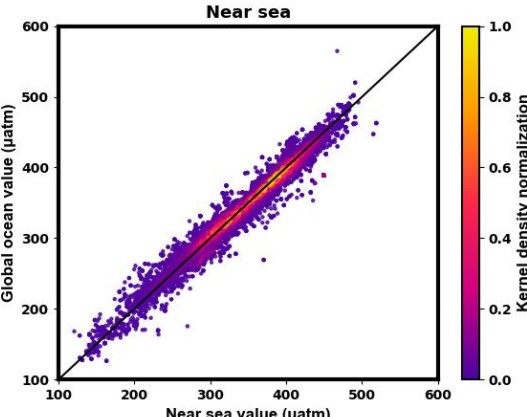

**Figure 13. Comparison of reconstruction accuracy in the near sea areas using different scale models, right axis: Normalized probability density of model residuals.**

### 3.5 Spatiotemporal analysis of surface pCO₂

At the global oceanic scale (Figure 14), the equatorial region experiences strong solar radiation and high temperatures, resulting in relatively low solubility of $CO_2$. Additionally, the presence of upwelling brings deep seawater rich in $CO_2$ to the surface, leading to an increase in surface $pCO_2$ concentration. Due to the low temperature environment in polar oceans, the solubility of $CO_2$ in seawater significantly increases. The sea ice coverage and strong wind fields in polar waters promote gas exchange between the atmosphere and the ocean, resulting in relatively low concentrations of $pCO_2$ on the sea surface. The surface $pCO_2$ in the Antarctic region is generally higher than that in the Arctic region, because the circulation system transports a large amount of seawater with high surface $pCO_2$ from low latitudes to high latitudes. At the same time, the melting and formation of sea ice also have an important impact on the distribution of surface $pCO_2$. Due to the wider coverage of sea ice, the Arctic region is less affected by the North Atlantic warm current, and its surface $pCO_2$ concentration is lower compared to the Antarctic region. In terms of time, the global ocean surface $pCO_2$ shows a trend of increasing year by year, which is related to global warming. The rising sea temperature in mid latitude waters leads to a decrease in $CO_2$ solubility and promotes an increase in surface $pCO_2$ concentration.





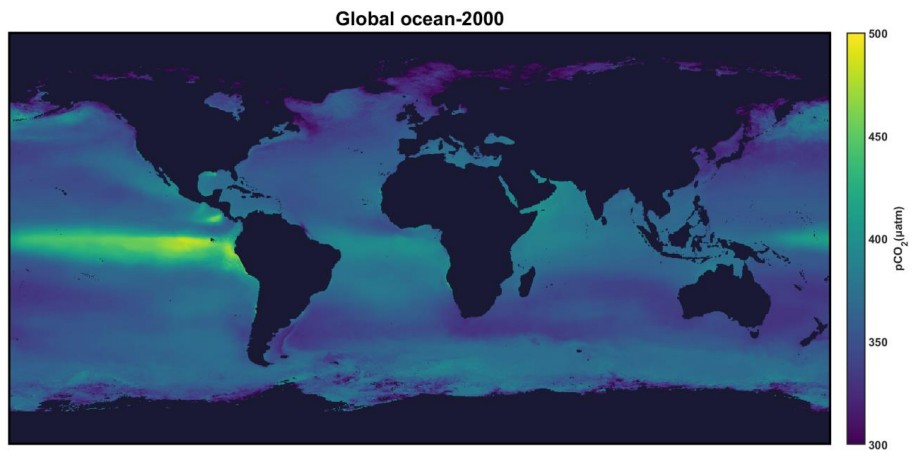

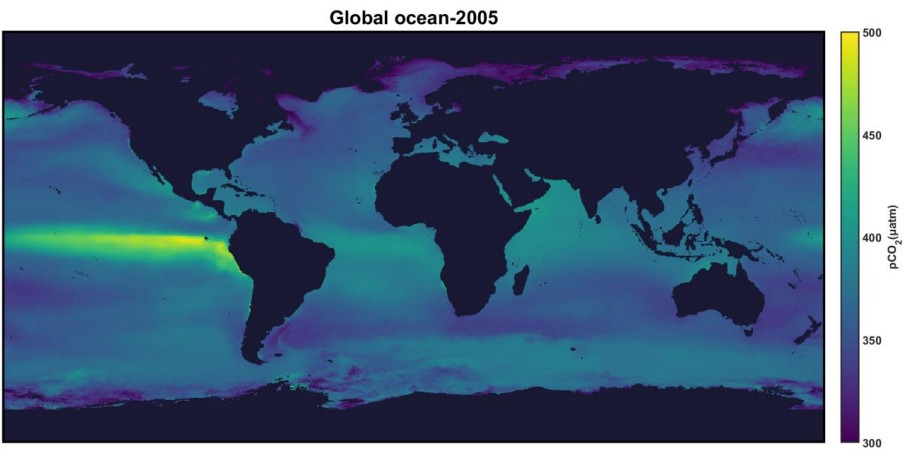

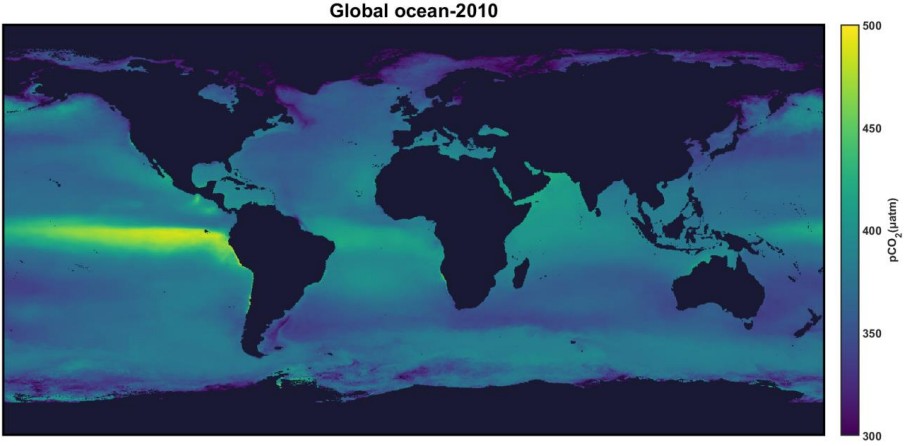

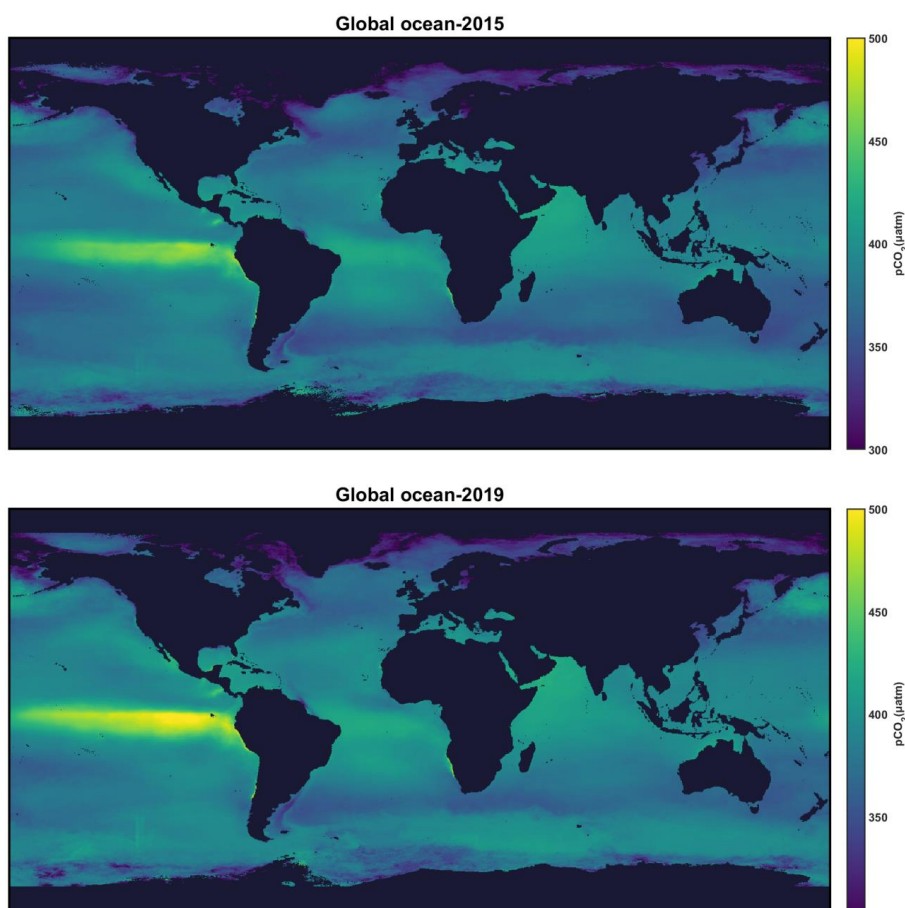

**Figure 14. Annual spatiotemporal variations of surface ocean pCO₂ in the globle ocean**

In the far sea areas (Figure 15), the surface pCO₂ is higher in the low latitude areas near the equator, particularly in the eastern equatorial Pacific. Mainly due to the upwelling of seawater in the region, which brings cold water rich in $CO_2$ from deep layers to the surface of the ocean, resulting in an increase in pCO₂ concentration on the sea surface. In the mid to high latitudes of the far sea region, the surface pCO₂ shows a low characteristic, which is due to the ocean circulation pattern promoting the mixing of surface seawater and deep seawater, resulting in relatively low surface pCO₂ concentration. The low temperature and strong biological pumping effect enhance the absorption of atmospheric $CO_2$ by the ocean, leading to a low surface pCO₂ concentration. In terms of time, the surface pCO₂ shows a trend of increasing year by year, especially after 2015. This is closely related to global climate change, changes in ocean circulation patterns, and the impact of human activities.



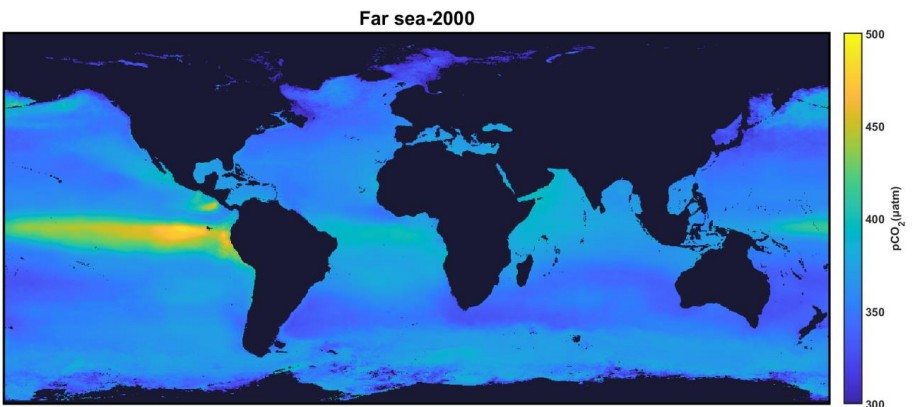

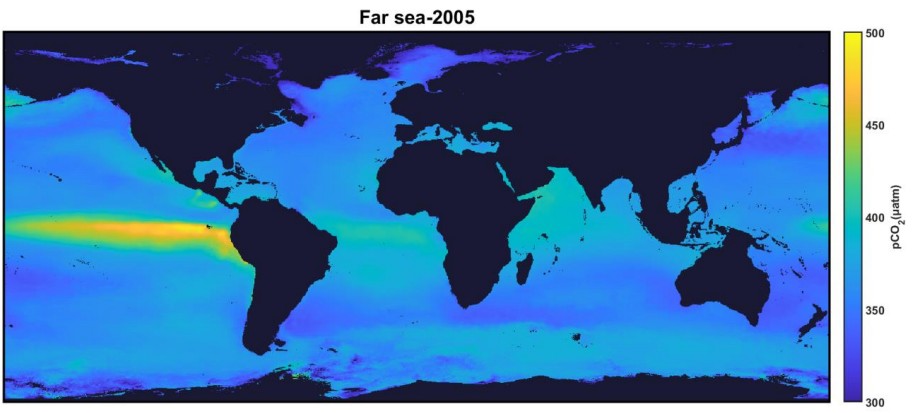

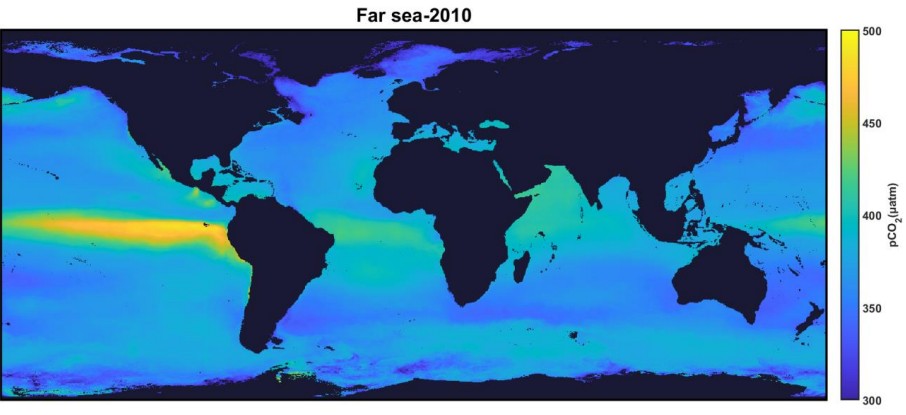



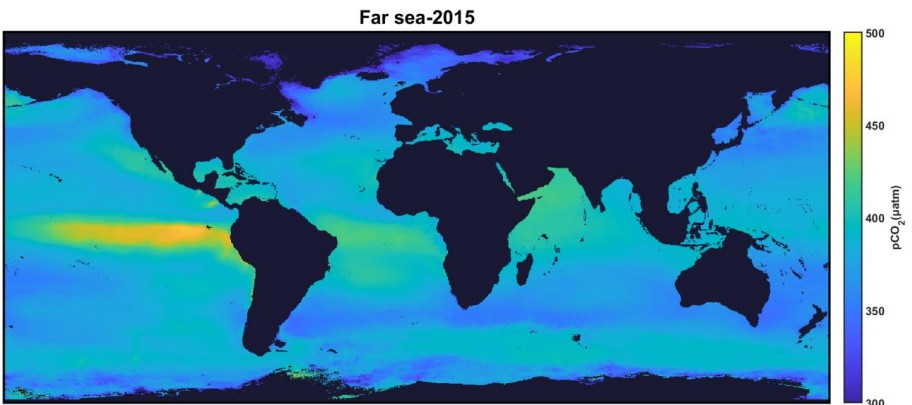

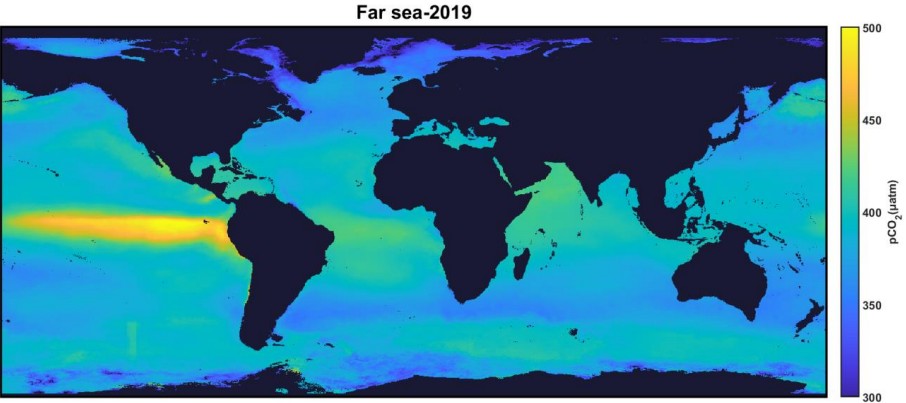

**Figure 15. Annual spatiotemporal variations of surface ocean $pCO_2$ in the far sea areas**

The exchange of $CO_2$ between seawater and atmosphere is frequent, and the surface $pCO_2$ value is relatively high. In mid to high latitude oceans, low-temperature seawater, polar cold water sinking, and deep seawater upwelling result in relatively low concentrations of $pCO_2$. The reconstruction results of surface $pCO_2$ in the near sea area (Figure 16) show that the equatorial region has strong solar radiation, high temperature seawater, and the influence of tropical cyclones and trade winds.The distribution characteristics of surface $pCO_2$ are significant along the eastern coast of Asia in the mid latitude region of the Northern Hemisphere. The surface $pCO_2$ in the Yellow Sea and Bohai Sea oceans is significantly lower than that in the coastal areas of eastern North America, which is related to the East Asian monsoon circulation and complex marine ecosystems. The surface $pCO_2$ in the border waters between Southeast Asia, the Indian Peninsula, North America, and South America is relatively high. Due to the influence of monsoon climate and tropical cyclones, high sea temperatures, as well as marine pollution caused by human activities, have collectively led to an increase in surface $pCO_2$. Temporally, the surface $pCO_2$ in near sea areas has been increasing year by year. Due to the increase in temperature in low latitude sea areas, the solubility of $CO_2$ in seawater decreases, and the upward trend of surface $pCO_2$ is more pronounced.




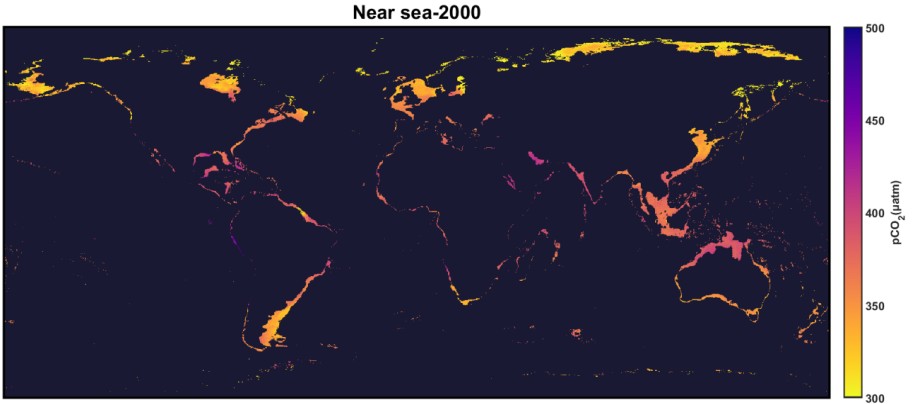

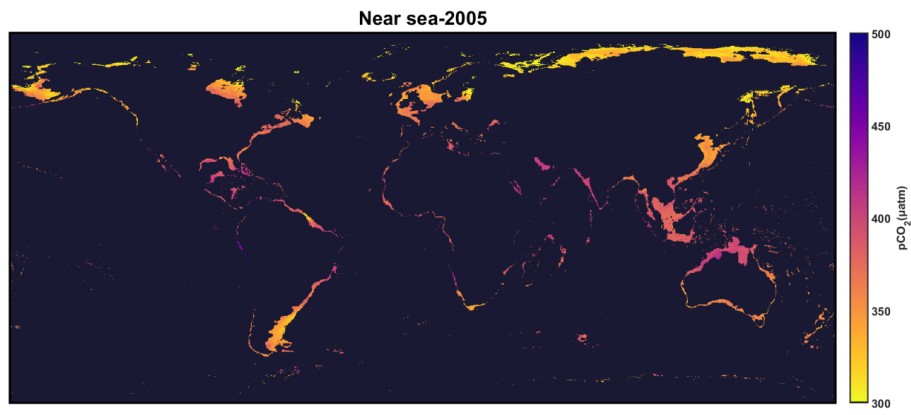

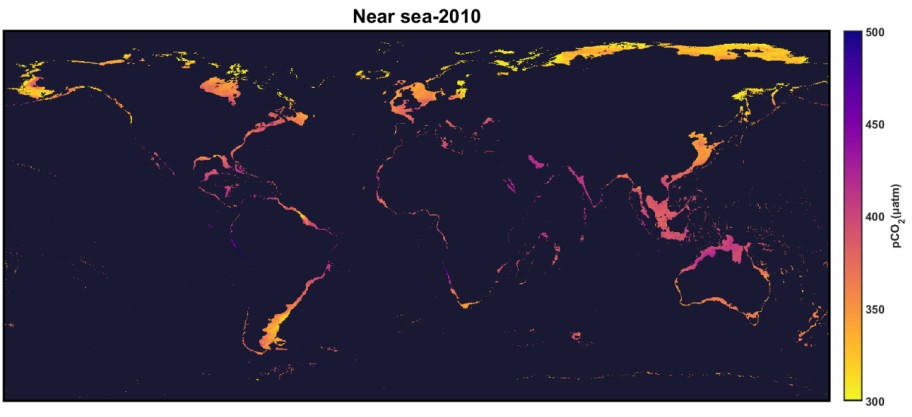



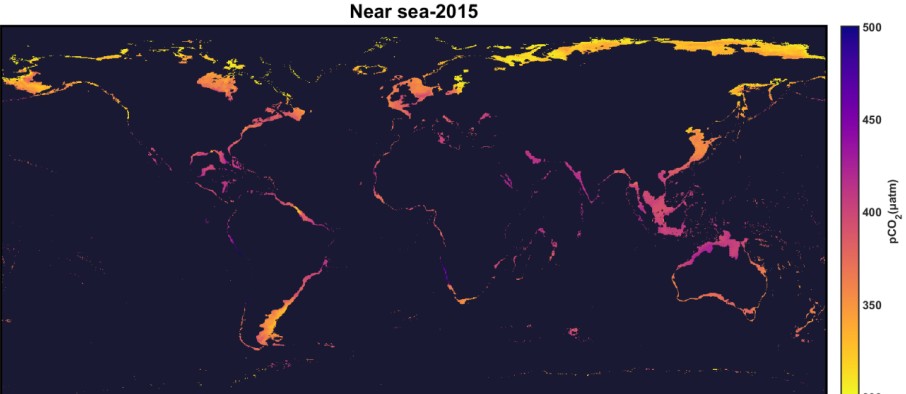

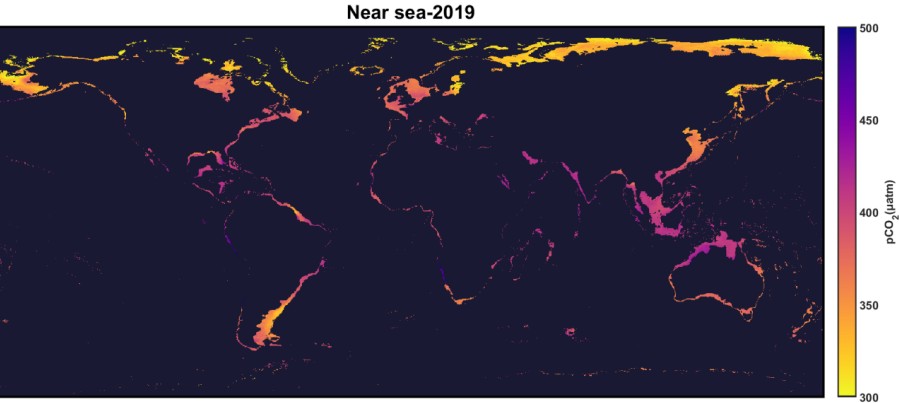

**Figure 16. Annual spatiotemporal variations of surface ocean pCO₂ in the near sea areas**

This study is based on a multi-scale analysis framework of the global ocean, far sea areas, and near sea areas. Using LDEO measured data combined with multi-source data, multiple machine learning models were used to construct and reconstruct the annual surface pCO₂ distribution of 0.25°× 0.25°from 2000 to 2019, revealing its spatiotemporal variation patterns and driving mechanisms. The research results indicate that the Random Forest (RF) model exhibits optimal performance at different scales and can effectively capture the spatiotemporal distribution characteristics of surface pCO₂. The distribution pattern of surface pCO₂ shows a pattern of "high at the equator and low at the poles" in space, and an increasing trend year by year in time. Different oceans exhibit different characteristics of changes due to the combined effects of natural factors and human activities. The acidity and alkalinity of seawater are the main driving factors for changes in surface pCO₂, and the contributions of other influencing factors vary at different scales.

Although this study has achieved certain results, the complexity of ocean carbon sinks still needs further exploration. Future research can focus on optimizing models, developing hybrid models, and combining advanced algorithms with ocean mechanism models; At the same time, we will strengthen interdisciplinary studies such as oceanography, ecology, and climatology to comprehensively reveal the process of ocean carbon cycling and



provide scientific basis for addressing climate change.

**Author contribution**

Conceptualization, **[H.W.]**; methodology, **[H.W.]** and **[Y.J.]**; software, **[X.L.]** and **[Y.J.]**; validation,

**[W .Z.]**, **[L.C.]**, and **[L.W.]**; formal analysis, **[Y.J.]**; investigation, **[W .Z.]**, **[L.W.]** and **[L.C.]**; resources, **[X.L.]**

and **[Y.J.]**; data curation, **[X.L.]** and **[Y.J.]**; writing—original draft preparation, **[Y.J.]**, **[Y.W.]** and **[M.L.]**;

writing—review and editing, **[H.W.]** and **[Z.L.]**; visualization, **[X.L.]** and **[L.C.]**; supervision **[H.W.]**; project

administration, **[H.W.]**; funding acquisition, **[H.W.]** All authors have read and agreed to the published version of

the manuscript.

**Competing interests**

The authors declare that they have no known competing financial interests or personal relationships that could
have appeared to influence the work reported in this paper.

**Financial support**

This research was funded by Key Laboratory of Land Satellite Remote Sensing Application, Ministry of Natural
Resources of the People' s Republic of China, grant numbers G202211, and the Ministry of Education
Industry-University Collaborative Education Project, grant numbers 220504039151258, and the Fundamental
Research Funds for the Central Universities, grant numbers 18CX02064A.

**Data availability**

Data will be made available on request.

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
