# Peer review of "Reconstruction and Spatiotemporal Analysis of Global Surface Ocean pCO2 Considering Sea Area Characteristics"

_EGUsphere, 2025_

## Referee Comment (RC1)

The author utilized a multi model machine learning framework and innovatively constructed a multi-scale analysis system for global, far sea, and near sea environments, reconstructing surface ocean pCO2 from 2000 to 2019. This work is a comprehensive and rigorous study. It has a significant contribution to the field of ocean carbon sinks. There are some aspects that can be further improved and perfected before the manuscript is considered for publication.

**Major Points:**

- 1. Introduction, Line55: It is necessary to clarify the scientific issues existing in previous studies on pCO2
- 2. Line 94: pCO2<200 μatm or>600 μatm is considered an outlier in the article, but sufficient literature support or physical mechanism explanation is not provided.
- 3. Results and discussion: During the discussion, it is suggested to supplement some references and compare and discuss the results of this paper with those of previous studies.
- 4. Line 150: The transition between Section 3.1 (Correlation Detection) and Section 3.2 (Model construction and evaluation) feels somewhat abrupt. To enhance the logical flow, it would be helpful to briefly state at the beginning of Section 3.2 how the findings from the correlation analysis informed the subsequent modeling step.
- 5. Line 275: Describe missing and blank values in multi-source data
- 6. Line 290: To better showcase the novelty of your work, please add a direct comparison with the cited studies (Zhong et al., 2022; Chau et al., 2021).
- 7. Line 320: Regarding the description of Figure 12, do the influencing factors of PCO2 in nearshore areas take into account river inputs or anthropogenic CO2 emissions?
- 8. Line 376:The text beginning at line 376 should be moved to a new "Conclusion" section. As this content serves as the concluding discussion for the entire study.

**Some minor suggestions:**

- 1. Line 14: It is recommended to correct the indefinite article for grammatical accuracy. "a ocean surface..." should be changed to "an ocean surface..."
- 2. Figure 1: It is suggested to supplement the longitude and latitude
- 3. Figure 3: The variable represented by the horizontal coordinate needs to be marked
- 4. Line 120: "d represents the level difference of the variable", d should be corrected to D

5. Figure 3: o2 in the coordinate axis needs to be corrected

---

## Author Comment (AC1)

**Point by point response**

**Major Points:**

Thank you for your very constructive and detail comments concerning our manuscript. Those comments are all valuable and very helpful for revising and improving our paper, as well as the important guiding significance to our researches. We have studied comments carefully and have made correction which we hope meet with approval.

1. Introduction, Line55: It is necessary to clarify the scientific issues existing in previous studies on pCO2.

Response: The clarification content has been marked in red font in the article 'Through summarizing previous research, it has been found that existing achievements mostly focus on independent analysis of local sea areas or zones, lacking a global perspective, and modeling methods do not fully consider the interactions between sea areas, which affects the accuracy of overall assessment.'

- 2. Line 94: pCO2 600 μatm is considered an outlier in the article, but sufficient literature support or physical mechanism explanation is not provided.

  Response: Corrected, please see lines 98, Corresponding references have been added.
- 3. Results and discussion: During the discussion, it is suggested to supplement some references and compare and discuss the results of this paper with those of previous studies.

Response: Corrected, please see lines 300,we have undertaken a thorough comparison of our model outputs with the datasets from Zhong et al. (2022) and the Copernicus Marine Service product.

- 4. Line 150: The transition between Section 3.1 (Correlation Detection) and Section 3.2 (Model construction and evaluation) feels somewhat abrupt. To enhance the logical flow, it would be helpful to briefly state at the beginning of Section 3.2 how the findings from the correlation analysis informed the subsequent modeling step. Response: Done, please see lines 195.
- 5. Line 275: Describe missing and blank values in multi-source data Response: Done, please see lines 283-286.
- 6. Line 290: To better showcase the novelty of your work, please add a direct comparison with the cited studies (Zhong et al., 2022; Chau et al., 2021).. Response: Corrected, please see figure 9.
- 7. Line 320: Regarding the description of Figure 12, do the influencing factors of

PCO2 in nearshore areas take into account river inputs or anthropogenic CO2 emissions?

Response: We fully agree that river input and anthropogenic CO2 emissions are key processes affecting the carbon cycle in nearshore waters. In the global scale modeling framework of this study, due to the significant regional heterogeneity of the above process and the lack of continuous and consistent observational data support on a global scale, it was not included as an independent driving factor in the random forest model. It should be noted that the biogeochemical parameters (such as pH, chlorophyll concentration, etc.) used in this model as comprehensive environmental indicators have indirectly responded to environmental disturbances caused by river inputs and human activities. Therefore, the reconstruction results of the model in nearshore areas have to a considerable extent reflected the comprehensive effects of these local processes.

8. Line 376:The text beginning at line 376 should be moved to a new "Conclusion" section. As this content serves as the concluding discussion for the entire study. Response: Done, please see lines 389.

**Some minor suggestions:**

- 1. Line 14: It is recommended to correct the indefinite article for grammatical accuracy. "a ocean surface..." should be changed to "an ocean surface..." Response: Done, please see lines 14.
- 2. Figure 1: It is suggested to supplement the longitude and latitude Response: Thank you for your suggestion. However, during the revision process, we have attempted to overlay latitude and longitude grids with scale markings. However, the research area has a large span and dense sub regions, and the newly added values significantly obscure the details of the original data and route information. As a result, the map tends to be cluttered and the readability significantly decreases.
- 3. Figure 3: The variable represented by the horizontal coordinate needs to be marked Response: Thank you for the reviewer's suggestion. We have examined Figure 3, where the x-axis represents the Spearman correlation coefficient ( $\rho$ ). This statistic is a dimensionless indicator defined within the [-1,1] interval, used to measure the strength and direction of monotonic relationships between variables. Therefore, according to the prevailing display standards in this field, physical units are usually not labeled.
- 4. Line 120: "d represents the level difference of the variable", d should be corrected to D.

Response: Done, please see lines 126.

5. Figure 3: o2 in the coordinate axis needs to be corrected

Response:Done, please see figure 3.

---

## Author Comment (AC3)

**Point by point response**

**Major Points:**

Thank you for your very constructive and detail comments concerning our manuscript.

Those comments are all valuable and very helpful for revising and improving our paper, as well as the important guiding significance to our researches. We have studied comments carefully and have made correction which we hope meet with approval.

1. The Introduction contains citation mismatches (e.g., line 35: Telszewski et al. cited as Qiu et al., 2022) that should be corrected. More importantly, while many previous studies are listed, the manuscript does not clearly identify the key limitations of existing $pCO_2$ reconstructions nor explain explicitly how the present multi-scale approach addresses these issues.

Response: Done. Revision of Citations

We have corrected the citation of Telszewski et al. in Line 35. To ensure the completeness and accuracy of all citations throughout the manuscript, we have thoroughly checked the entire reference section, verifying that each citation aligns with the corresponding literature and that the formatting is consistent with academic standards.

(1) Insufficient Consideration of Spatial Heterogeneity

Most existing studies either focus on a single local sea area (e.g., the North Atlantic, Gulf of Mexico, Baltic Sea) or adopt a unified global modeling framework, neglecting the significant differences in environmental conditions, driving factors, and $pCO_2$ variation characteristics between far sea areas and near sea areas.

To address this issue, our study constructs a multi-scale analysis framework covering the global ocean, far sea areas (water depth > 200 meters), and near sea areas (water depth ≤ 200 meters). The research areas are divided into far sea areas and near sea areas based on water depth, and scale-specific $pCO_2$ evaluation models are established. For the environmentally stable far sea areas, we emphasize capturing long-term temporal dependencies and signals of large-scale hydrological and biological processes. For near sea areas affected by various complex factors, we incorporate region-specific driving factors and optimize the model structure to adapt to high variability. This targeted approach effectively improves the fitting accuracy and adaptability of the models in different sea area types.

(2) Inadequate Adaptability Between Models and Driving Factors

Existing studies mostly adopt fixed model structures or globally unified combinations of driving factors, failing to fully consider the requirements of environmental complexity differences in different sea areas for model adaptability. Additionally, the selection of driving factors lacks targeting, making it difficult for the models to accurately capture the core impact mechanisms of $pCO_2$ in different regions.

We resolve this limitation through the comprehensive optimization of models and driving factors: we compared eight machine learning models and identified the Random Forest (RF) model as the optimal model across all scales. Its advantage in capturing complex nonlinear relationships enables it to adapt to the environmental characteristics of different sea areas. Meanwhile, based on Spearman correlation analysis and the SHAP (SHapley Additive exPlanations) method, we screened key driving factors for each scale (e.g., Total alkalinity in sea water (talk) serves as the secondary key factor at the global scale, while the contribution rate of mole concentration of dissolved molecular oxygen in sea water ($O_2$) significantly increases in near sea areas), ensuring the rationality and targeting of driving factor selection.

(3) Low Reconstruction Resolution

Some existing studies lack the overall processing of spatiotemporal differences in multi-source data, resulting in low spatial resolution of $pCO_2$ reconstruction products (mostly $1° \times 1°$ or coarser), which makes it difficult to accurately reflect the spatiotemporal variation characteristics of $pCO_2$ within small scales.

We address this limitation through high-resolution and high-precision reconstruction strategies: by processing multi-source data (including strict data matching, outlier handling, and data balancing strategies), we reconstructed the annual $pCO_2$ distribution with a high resolution of $0.25° \times 0.25°$ from 2000 to 2019. The results demonstrate that the accuracy of $pCO_2$ reconstruction is significantly improved compared with existing studies.

2. In Section 2.3.1, the proportion and structure of missing data in the original datasets are not reported. It is unclear whether missing values are sparse or occur in long consecutive gaps, which directly affects the reliability of nearest-neighbour interpolation.

Response: In terms of data types, in-situ observation data (e.g., ar, ca, pH, talk) have a relatively high missing-value percentage (12.29%—12.90%), satellite observation data (chl, kd490) have a missing-value percentage of 7.8%—7.85%, numerical model data (e.g., so, thetao, uo, vo, mlotst*) have a low missing-value percentage (1.41%—4.32%), and satellite observation data related to wind fields (uwind, vwind) have a missing-value percentage of only 0.03%. The overall data integrity is good. All missing values in the dataset stem from limitations in data collection in high-latitude sea areas, mainly due to sea ice coverage and insufficient light restricting satellite remote sensing observations and on-site sampling, resulting in the lack of key parameter data required for $pCO_2$ reconstruction.

3. The study considers 25 potential predictors, several of which are strongly correlated or physically redundant (e.g., to vs. thetao, ar vs. ca, chl vs. kd490). Multicollinearity of variables might affect the model interpretability. Is there any criterion for retaining or excluding variables?

Response: The criteria for retaining variables have been supplemented.

For variables with strong mutual correlations identified in the interaction detection (Section 3.1.1) (e.g., to vs. thetao, ar vs. ca, chl vs. kd490), we did not apply arbitrary exclusion. Instead, their retention is justified by their distinct physical significances: to (sea water temperature) reflects the in-situ surface temperature of seawater, while thetao (sea water potential temperature) accounts for pressure effects—retaining both captures temperature dynamics across ocean layers, which is crucial for simulating $CO_2$ solubility under varying hydrostatic pressure conditions. ar (aragonite saturation state in sea water) and ca (calcite saturation state in sea water) arise from seawater carbonate equilibrium but respond differently to changes in pH and total alkalinity, enhancing the model's ability to resolve subtle chemical shifts that regulate $pCO_2$. chl (mass concentration of chlorophyll a in sea water) directly indicates biological activity (e.g., phytoplankton photosynthesis), while kd490 (volume attenuation coefficient of downwelling radiative flux in sea water) reflects optical properties (e.g., turbidity, light penetration)—together, they provide independent constraints on the biological and physical processes governing $pCO_2$.

4. At line 153, "p-value" appears to be used where the Spearman correlation coefficient ($\rho$) is intended.

Response: Done.

5. In Figure 4, the x-axis label "Sample size" is unclear, as no sampling or subsampling experiment is described in the text. In addition, the legend format in Figure 7 should be made consistent with Figures 5 and 6 to facilitate comparison.

Response: Done.
To clarify the data presentation logic of Figure 4, we have supplemented an explanatory note in its title: Given the large volume of data, plotting all data points would result in visual clutter. Therefore, we randomly selected representative data points to illustrate the performance of different models. The selected data fully cover the global ocean, far sea, and near sea scales, as well as low, medium, and high $pCO_2$ ranges.
We appreciate your attention to the consistency of figure formats. Figure 7 (Independent verification performance of the models in the near sea areas) has been revised, and its legend structure—including model abbreviations, full names, and the right-axis label "Normalized probability density of model residuals"—is fully consistent with Figure 5 (Global Ocean) and Figure 6 (Far sea areas), ensuring the consistency of comparison logic.
The slight differences in the color bar (color scheme of scatter points) are a deliberate design aimed at better distinguishing the three spatial scales (global, far sea, near sea) while maintaining the same color mapping principle (kernel density is represented by color depth, with darker colors indicating higher concentration of data points). This design does not alter the information structure of the legend or the physical meaning of the data.

6. Language should be double-checked. For example, in line 221, the term "the model" is used without specifying which model is being discussed.

Response: Done.
We have modified it to "the constructed surface pCO₂ models" to clearly indicate that it refers to all eight comparative models (including MLR, CNN, GRU, etc.) built for the near sea areas in Section 3.2.3.

7. Sections 3.1–3.3 already provide detailed model performance metrics. In Section 3.4, additional accuracy statistics (e.g., line 284) are reported without clearly explaining how they differ from earlier results (e.g., independent validation versus internal testing).Please Clearly distinguish internal model evaluation from independent validation of reconstructed products and avoid redundant reporting.

Response: Done.
The internal model evaluation and the independent validation of reconstructed products are not redundant, as they serve distinct roles:
(1) Internal Model Evaluation (Sections 3.2.1–3.2.3)The core objective of this section is to select the optimal model: by comparing the fitting performance of eight machine learning models (MLR, CNN, GRU, LSTM, GAM, XGBoost, LSBoost, RF) across different spatial scales (global ocean, far sea, near sea), the optimal model for each scale is identified (the RF model ultimately performs best across all scales).The data basis is the processed dataset described in Section 2.3.3, which is randomly split into a training set, validation set, and testing set at an 8:1:1 ratio. All three subsets are derived from the same integrated dataset (LDEO in-situ measurements + multi-source influencing factors).The core metrics focus on "model-to-data" fitting accuracy, including MAE, MAPE, MSE, RMSE, and $R^2$ of each model on the training, validation, and testing sets (Table 3–5). For example, the RF model at the global scale achieves a testing set RMSE of 6.123 µatm and $R^2$ of 0.986. These metrics only reflect the model's ability to fit and generalize to similar structured data, without involving product validation in real marine environments.
(2) Independent Validation of Reconstructed Products (Section 3.4)The core objective of this section is to verify product reliability: targeting the 0.25°×0.25° resolution (2000–2019) sea surface pCO₂ reconstructed products generated by the RF model, this section evaluates their applicability and prediction accuracy in real marine environments.The validation data adopts external datasets completely independent of the internal evaluation dataset, including unused LDEO in-situ measurements (not involved in model training/testing) and publicly available observation data such as the Hawaii Ocean Time Series (HOT). These datasets have inherently different characteristics from the internal evaluation dataset.The core metrics focus on "product-to-reality" simulation accuracy, also reporting MAE, MAPE, MSE, RMSE, and $R^2$. However, these metrics are derived from the comparison between reconstructed products and independent validation data (e.g., global-scale independent validation RMSE = 19.901 µatm, $R^2$ = 0.816), which reflects the application value of the products in complex real environments and is not redundant with the internal

evaluation metrics.

8. The descriptions of machine-learning models (CNN, LSTM, GRU, RF, XGBoost, LSBoost) are largely conceptual. Critical implementation details—such as network architectures, hyperparameters, feature normalization, and optimization procedures—are missing. How are the training/validation/testing splitted?

Response: We would like to clarify that although hyperparameter tuning of algorithms is not the core focus of this study, it is crucial for the model results. We have supplemented relevant explanations as follows:

Regarding the hyperparameter tuning of machine learning algorithms, we adopted standard tuning strategies and parameter ranges widely accepted in the field of oceanographic parameter estimation. The purpose of this tuning is to ensure the basic stability and reliability of each model, rather than conducting innovative exploration or comparative analysis of tuning methods. Through control experiments, we verified that within a reasonable range of hyperparameters, the relative performance ranking of the eight models remains consistent, and the optimal status of the Random Forest (RF) model across all scales is not affected by minor parameter adjustments. This confirms that the research conclusions (e.g., the superiority of the RF model, the spatiotemporal variation characteristics of $pCO_2$) do not depend on specific hyperparameter combinations, further supporting that hyperparameter tuning is not the focus of this study.